# PERPLEXITY-TRAP: PLM-BASED RETRIEVERS OVERRATE LOW PERPLEXITY DOCUMENTS

**Haoyu Wang**[1]*, **Sunhao Dai**[1]*, **Haiyuan Zhao**[1], **Liang Pang**[2], **Xiao Zhang**[1]
**Gang Wang**[3], **Zhenhua Dong** [3], **Jun Xu**[1]†, **Ji-Rong Wen**[1]
[1]Gaoling School of Artificial Intelligence, Renmin University of China, Beijing, China
[2]CAS Key Laboratory of AI Safety, Institute of Computing Technology, Beijing, China
[3]Huawei Noah's Ark Lab, Shenzhen, China
{wanghaoyu0924,sunhaodai,junxu}@ruc.edu.cn,

## ABSTRACT

Previous studies have found that PLM-based retrieval models exhibit a preference for LLM-generated content, assigning higher relevance scores to these documents even when their semantic quality is comparable to human-written ones. This phenomenon, known as source bias, threatens the sustainable development of the information access ecosystem. However, the underlying causes of source bias remain unexplored. In this paper, we explain the process of information retrieval with a causal graph and discover that PLM-based retrievers learn perplexity features for relevance estimation, causing source bias by ranking the documents with low perplexity higher. Theoretical analysis further reveals that the phenomenon stems from the positive correlation between the gradients of the loss functions in language modeling task and retrieval task. Based on the analysis, a causal-inspired inference-time debiasing method is proposed, called **C**ausal **D**iagnosis and **C**orrection (CDC). CDC first diagnoses the bias effect of the perplexity and then separates the bias effect from the overall estimated relevance score. Experimental results across three domains demonstrate the superior debiasing effectiveness of CDC, emphasizing the validity of our proposed explanatory framework [1].

## 1 INTRODUCTION

The rapid advancement of large language models (LLMs) has driven a significant increase in AI-generated content (AIGC), leading to information retrieval (IR) systems that now index both human-written and LLM-generated contents (Cao et al., 2023; Dai et al., 2024b; 2025). However, recent studies (Dai et al., 2024a;c; Xu et al., 2024) have uncovered that Pretrained Language Model (PLM) based retrievers (Guo et al., 2022; Zhao et al., 2024) exhibit preferences for LLM-generated documents, ranking them higher even when their semantic quality is comparable to human-written content. This phenomenon, referred to as **source bias**, is prevalent among various popular PLM-based retrievers across different domains (Dai et al., 2024a). If the problem is not resolved promptly, human authors' creative willingness will be severely reduced, and the existing content ecosystem may collapse. So it's urgent to comprehensively understand the mechanism behind source bias, especially when the amount of online AIGC is rapidly increasing (Burtch et al., 2024; Liu et al., 2024).

Existing studies identify perplexity (PPL) as a key indicator for distinguishing between LLM-generated and human-written contents (Mitchell et al., 2023; Bao et al., 2023). Dai et al. (2024c) find that although the semantics of the text remain unchanged, LLM-rewritten documents possess much lower perplexity than their human-written counterparts. However, it's still unclear *whether document perplexity has a causal impact on the relevance score estimation of PLM-based retrievers* (which may lead to source bias), and *if so, why such causal impact exists*.

In this paper, we delve deeper into the cause of source bias by examining the role of perplexity in PLM-based retrievers. By manipulating sampling temperature when generating with LLMs, we

---

*Equal contributions.
†Corresponding author.

[1]Codes are available at https://github.com/WhyDwelledOnAi/Perplexity-Trap.

observe a negative correlation between estimated relevance scores and perplexity. Inspired by this, we construct a causal graph where document perplexity plays as a treatment and document semantic plays as a confounder (Figure 2). We adopt a two-stage least squares (2SLS) regression procedure (Angrist and Pischke, 2009; Hartford et al., 2017) to eliminate the influence of confounders when estimating this biased effect, the experimental results indicate the effect is significantly negative. Based on these findings, the cause of source bias can be elucidated as the unexpected causal effect of perplexity on estimated relevance scores. For semantically identical documents, the documents with low perplexity causally get higher estimated relevance scores from PLM-based retrievers. Since LLM-generated documents typically have lower perplexity than human-written ones, they receive higher estimated relevance scores and are ranked higher, leading to the presence of source bias.

To further understand why estimated relevance scores of PLM-based retrievers are influenced by perplexity, we provide a theoretical analysis for the overlap between masked language modeling (MLM) task and mean-pooling retrieval task. Analysis in the linear decoder scenario shows that, the retrieval objective's gradients are positively correlated to the language modeling gradients. This correlation causes the retrievers to consider not only the document semantics required for retrieval but also the bias introduced by perplexity. Meanwhile, this correlation further explains the trade-off between retrieval performance and source bias observed in previous study (Dai et al., 2024a): the stronger the ranking performance of the PLM-based retrievers, the greater the impact of perplexity.

Based on the analysis, we propose an inference-time debiasing method called **CDC** (**C**ausal **D**iagnosis and **C**orrection). With the proposed causal graph, we separate the causal effect of perplexity from the overall estimated relevance scores during inference, achieving calibrated unbiased relevance scores. Specifically, CDC first estimates the biased causal effect of perplexity on a small set of training samples, which is then applied to de-bias the test samples at the inference stage. This debiasing process is inference-time and can be seamlessly integrated into existing trained PLM-based retrievers. We demonstrate the debiasing effectiveness of CDC with experiments across six popular PLM-based retrievers. Experimental results show that the estimated causal effect of perplexity can be generalized to other data domains and LLMs, highlighting its practical potential in eliminating source bias.

We summarize the major contributions of this paper as follows:

• We construct a causal graph and estimate the causal effect through experiments, demonstrating that PLM-based retrievers causally assign higher relevance scores to documents with lower perplexity, which is the cause of source bias.

• We provide a theoretical analysis explaining that the effect of perplexity in PLM-based retrievers is due to the positive correlation between objective gradients of retrieval and language modeling.

• We propose CDC for PLM-based retrievers to counteract the biased effect of perplexity, with experiments demonstrating its effectiveness and generalizability in eliminating source bias.

## 2 RELATED WORK

With the rapid development of LLMs (Zhao et al., 2023), the internet has quickly integrated a huge amount of AIGC (Cao et al., 2023; Dai et al., 2024b; 2025). Potential bias may occur when these generated contents are judged by neural networks as a competitor together with human works. For example, Dai et al. (2024c) are the first to highlight a paradigm shift in information retrieval (IR): the content indexed by IR systems is transitioning from exclusively human-written corpora to a coexistence of human-written and LLM-generated corpora. They then uncover an important finding that mainstream neural retrievers based on pretrained language models (PLMs) prefer LLM-generated content, a phenomenon termed source bias (Dai et al., 2024a;c). Xu et al. (2024) further discover that this bias extends to text-image retrieval, and similarly, other works further observe the existence of source bias in other IR scenarios, such as recommender systems (RS) (Zhou et al., 2024), retrieval-augmented generation (RAG) (Chen et al., 2024) and question answering (QA) (Tan et al., 2024). In the context of LLMs-as-judges, similar bias is discovered as self-enhancement bias (Zheng et al., 2024), likelihood bias (Ohi et al., 2024), and familiarity bias (Stureborg et al., 2024), where LLM overates AIGC when serving as a judge.

Existing works provide intuitive explanations suggesting that this kind of bias may stem from coupling between neural judges and LLMs (Dai et al., 2024c; Xu et al., 2024), such as similarities in model

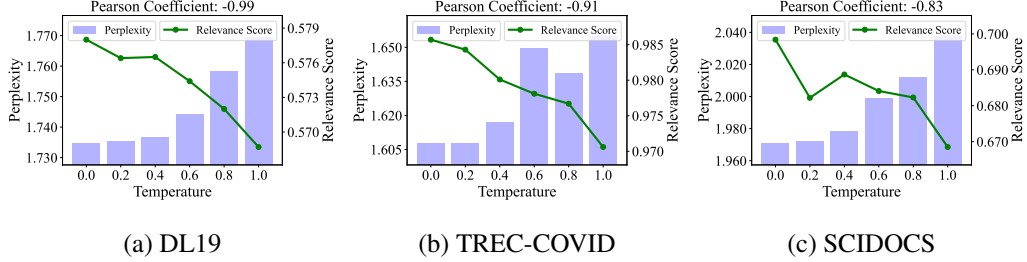

(a) DL19                    (b) TREC-COVID                    (c) SCIDOCS

Figure 1: Perplexity and estimated relevance scores of ANCE on positive query-document pairs in three dataset, where documents are generated by LLM rewriting with different sampling temperatures. The Pearson coefficients highlight the significant negative correlation between the two variables.

architectures and training objectives. However, the specific nature of this coupling, how it operates to cause source bias, and why it exists remains unclear. Ohi et al. (2024) find the correlation between perplexity and bias, while our work is the first to systematically analyze the effect of perplexity for neural models' preference. Given that both PLMs and LLMs are highly complex neural network models, investigating this question is particularly challenging and difficult.

## 3 ELUCIDATING SOURCE BIAS WITH CAUSAL GRAPH

This section first conducts intervention experiments to illustrate the motivation. Subsequently, we construct a causal graph to explain source bias and demonstrate the rationality of the causal graph.

### 3.1 MOTIVATION: INTERVENTION EXPERIMENTS ON TEMPERATURE

Previous studies have revealed a significant difference in the perplexity (PPL) distribution between LLM-generated content and human-written content (Mitchell et al., 2023; Bao et al., 2023), suggesting that PPL might be a key indicator for analyzing the cause of source bias (Dai et al., 2024c). To verify whether perplexity causally affects estimated relevance scores, we use LLMs (in following chapters the LLMs we use are Llama2-7B-chat (Touvron et al., 2023) unless emphasized) to generate documents with almost identical semantics but varying perplexity, where semantics are expected as the only associated variable when retrieval.

Specifically, we manipulate the sampling temperatures during generation to obtain LLM-generated documents with different PPLs but similar semantic content. Following the method of Dai et al. (2024c), we use the following simple prompt: "*Please rewrite the following text: {human-written text}*". We also recruit human annotators to conduct evaluations to ensure the quality of the generated LLM content. The results, shown in Appendix E.2.1, indicate that there are fewer quality discrepancies between documents generated at different sampling temperatures compared to the original human-written documents. This ensures the reliability of the subsequent experiments.

We then explore the relationship between perplexity and estimated relevance scores on the corpora generated with different temperatures, where perplexity is calculated by BERT masked language modelling following previous work (Dai et al., 2024c). Figure 1 presents the average perplexity and estimated relevance scores by ANCE across three datasets from different domains. As expected, lower sampling temperatures result in less randomness in LLM-generated content and thus lower PPL. Meanwhile, we find that documents generated with lower temperatures were also more likely to be assigned higher estimated relevance scores. The Pearson coefficients for the three datasets are all below -0.8, emphasizing the strong negative linear correlation between document perplexity and relevance score. Similar results for other PLM-based retrievers are provided in Appendix E.2.2.

Since document semantics remain unchanged during rewriting, the synchronous variation between document perplexity and estimated relevance scores reflects a causal effect. These findings offer an intuitive explanation for source bias: LLM-generated content typically has lower PPL, and since documents with lower perplexity are more likely to receive higher relevance scores, LLM-generated content is more likely to be ranked highly, leading to source bias.

## 3.2 CAUSAL GRAPH FOR SOURCE BIAS

Inspired by the findings above, we propose a causal graph to elucidate source bias (Fan et al., 2022), as illustrated in Figure 2. Let $\mathcal{Q}$ denotes the query set and $\mathcal{C}$ denote the corpus. During the inference stage for a certain PLM-based retriever, given a query $q \in \mathcal{Q}$ and a document $d \in \mathcal{C}$, the estimated relevance score $\hat{R}_{q,d} \in \mathcal{R}$ is simultaneously determined by both the golden relevance score $R_{q,d} \in \mathcal{R}$ and document perplexity $P_d \in \mathcal{R}_+$. Note that the fundamental goal of IR is to calculate the similarity between document semantics $M_d$ and query semantics $M_q$ for document ranking, $R_{q,d} \rightarrow \hat{R}_{q,d}$ is considered an unbiased effect, while the influence of $P_d \rightarrow \hat{R}_{q,d}$ is considered as a biased effect. Subsequently, we explain the rationale behind each edge in the causal graph as follows:

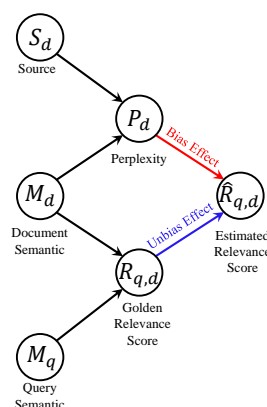

First, let the document source $S_d$ is a binary variable where $S_d = 1$ denotes the document is generated by LLM and $S_d = 0$ denotes the document is written by human. As suggested in (Dai et al., 2024c), LLM-generated documents through rewriting possess lower perplexity than their original documents, even though there is no significant difference in their semantic content. Thus, an edge $S_d \rightarrow P_d$ exists. This phenomenon can be attributed to two main reasons: (1) Sampling strategies aimed at probability maximization, such as greedy algorithms, discard long-tailed documents during LLM inference. More detailed analysis and verification can be found in (Dai et al., 2024c). (2) Approximation error during LLM training causes the tails of the document distribution to be lost (Shumailov et al., 2023).

Next, the document semantics $M_d$ reflect the topics of the document $d$, including domain, events, sentiment information, and so on. Since documents with different semantic meanings convey different amounts of information, their difficulties in masked token prediction vary. This means that different document semantics lead to different document perplexities. For example, colloquial conversations are more predictable than research papers due to their less specialized vocabulary. Thus, the content directly affects the perplexity, establishing the edge $M_d \rightarrow P_d$.

Figure 2: The proposed causal graph for explaining source bias.

Finally, as retrieval models are trained to estimate ground-truth relevance, their outputs are valid approximations of the golden relevance scores, making $M_d \rightarrow R_{q,d} \leftarrow M_q$ a natural unbiased effect. However, retrieval models may also learn non-causal features unrelated to semantic matching, especially high-dimensional features in deep learning. According to findings in Section 3.1, document perplexity $P_d$ has emerged as a potential non-causal feature learned by PLM-based retrievers, where higher relevance estimations coincide with lower document perplexity. Moreover, Since document perplexity is determined at the time of document generation, which temporally predates the existence of estimated relevance scores, document perplexity should be a cause rather than a consequence of changes in relevance. Hence, a biased effect of $P_d \rightarrow \hat{R}_{q,d}$ exists.

## 3.3 EXPLAINING SOURCE BIAS VIA THE PROPOSED CAUSAL GRAPH

Based on the causal graph constructed above, source bias can be explained as follows: Although the content generated by LLMs retains similar semantics to the human-written content, LLM-generated content typically exhibits lower perplexity. Coincidentally, retrievers learn and incorporate perplexity features into their relevance estimation processes, consequently assigning higher relevance scores to LLM-generated documents. This leads to the lower ranking of human-written documents.

It is worth noting that source bias is an inherent issue in PLM-based retrievers. Before the advent of LLMs, these retrievers had already learned non-causal perplexity features from purely human-written corpora. However, because the document ranking was predominantly conducted on human-written corpora, the relationship between PLM-based retrievers and perplexity was not evident. As powerful LLMs have become more accessible, the emergence of LLM-generated content has accentuated the perplexity effect. The content generated by LLMs exhibits a perceptibly different perplexity distribution compared to human-written content. This disparity in perplexity distribution causes documents from different sources to receive significantly different relevance rankings.

Table 1: Quantified causal effects (and corresponding $p$-value) for document perplexity on estimated relevance scores via two-stage regression. Bold indicates that the estimate can pass a significance test with $p$-value$< 0.05$. Significant negative causal effects are prevalent across various PLM-based retrievers in different domain datasets.

| Dataset | BERT | RoBERTa | ANCE | TAS-B | Contriever | coCondenser |
|---|---|---|---|---|---|---|
| DL19 | **-9.32 (1e-4)** | **-28.15 (2e-12)** | **-0.52 (9e-3)** | **-0.96 (1e-2)** | -0.02 (0.33) | **-0.69 (3e-2)** |
| TREC-COVID | **-1.69 (2e-2)** | 2.42 (8e-2) | 0.09 (0.21) | **-0.48 (6e-3)** | **-0.05 (7e-7)** | **-0.32 (8e-3)** |
| SCIDOCS | -2.44 (6e-2) | **-6.42 (2e-3)** | -0.23 (0.15) | -0.39 (0.10) | -0.02 (0.24) | -0.26 (0.41) |

## 4 EMPIRICAL AND THEORETICAL ANALYSIS ON THE EFFECT OF PERPLEXITY

In this section, we conduct empirical experiments and theoretical analysis to substantiate that PLM-based retrievers assign higher relevance scores to documents with lower perplexity.

### 4.1 EXPLORING THE BIASED EFFECT CAUSED BY PERPLEXITY

#### 4.1.1 ESTIMATION METHODS

From the temperature intervention experiments in Section 3.1, we observe a clear negative correlation between document perplexity and estimated relevance scores. Despite human evaluation allows us to largely confirm that document semantics $M_d$ generated from different temperatures are almost the same, estimating the biased effect of $P_d \rightarrow \hat{R}_{q,d}$ directly is problematic due to inevitable minor variations in document semantics, which, though subtle, are significant in causal effect estimation. From the causal view, to robustly estimate the causal effect of $P_d \rightarrow \hat{R}_{q,d}$, the document semantics $M_d$, query semantics $M_q$ and golden relevance scores $R_{q,d}$ are considered as confounders. Therefore, directly estimating this biased causal effect is not feasible without addressing this confounding factor.

We use 2SLS based on instrumental variable (IV) methods (Angrist and Pischke, 2009; Hartford et al., 2017) to more accurately evaluate the causal effect of document perplexity on estimated relevance scores, more details about the method can be found in Appendix D. According to the causal graph, document source $S_d$ serves as an IV for estimating the effect of $P_d \rightarrow \hat{R}_{q,d}$. The IV is independent of confounders: query semantics $M_q$, document semantics $M_d$, and golden relevance scores $R_{q,d}$.

In the first stage of the regression, we use linear regression to predict document perplexity $P_d$ based on document source $S_d$:

$$P_d = \beta_1 S_d + \tilde{P}_d, \tag{1}$$

where $\tilde{P}_d$ is independent with document source $S_d$ and therefore depends solely on document semantics $M_d$. As a result, we obtain coefficient $\hat{\beta}_1$ and the predicted document perplexity $\hat{P}_d$. In the second stage, we substitute $P_d$ with $\hat{P}_d = \hat{\beta}_1 S_d$ to estimate the predicted relevance score $\hat{R}_{q,d}$ from the certain PLM-based retrievers:

$$\hat{R}_{q,d} = \beta_2 \hat{P}_d + \tilde{R}_{q,d}, . \tag{2}$$

where residual term $\tilde{R}_{q,d}$ represents the part of the estimated relevance scores that can't be explained by document perplexity. Since $\hat{P}_d$ is independent of document semantics $M_d$, the estimated coefficient $\hat{\beta}_2$ can accurately reflect the causal effect of perplexity on estimated relevance scores.

#### 4.1.2 EXPERIMENTAL RESULTS AND ANALYSIS

In this section, we apply the causal effect estimation method described previously to assess the impact of document perplexity $P_d$ on the estimated relevance score $\hat{R}_{q,d}$.

**Models.** To comprehensively evaluate this causal effect, we select several representative PLM-based retrieval models from the Cocktail benchmark (Dai et al., 2024a), including: (1) BERT (Devlin et al., 2019); (2) RoBERTa (Liu et al., 2019); (3) ANCE (Xiong et al., 2020); (4) TAS-B (Hofstätter et al., 2021); (5) Contriever (Izacard et al., 2022); (6) coCondenser (Gao and Callan, 2022). We employ the officially released checkpoints. For more details, please refer to Appendix E.1.

**Datasets.** We select three widely-used IR datasets from different domains to ensure the broad applicability of our findings: (1) *DL19* dataset (Craswell et al., 2020) for exploring retrieval across miscellaneous domains. (2) *TREC-COVID* dataset (Voorhees et al., 2021) focused on biomedical information retrieval. (3) *SCIDOCS* (Cohan et al., 2020) dedicated to the retrieval of scientific scholarly articles. Given that source bias arises from the ranking orders of positive samples from different sources, we only compare the estimated relevance scores of human-written and LLM-generated relevant documents against their corresponding queries.

**Results and Analysis.** The results across different datasets and different PLM-based retrievers are shown in Table 1. As we can see, in most cases, perplexity exhibits a consistently negative causal effect on relevance estimation, with documents of lower perplexity more likely to receive higher relevance scores. Although this causal effect is relatively weak, it is statistically significant, with $p$-values $< 0.05$ in most instances. We also explore whether this causal effect changes with different sampling temperature. Results in Appendix Table 5 indicate that $\hat{\beta}_2$ is robust for temperature changes, that is, this causal effect is independent with generation temperature. This finding is crucial as retrieval tasks emphasize the relative ranking of relevance scores rather than their absolute values. Even a slight preferential increase in estimated relevance scores for LLM-generated content over human-written content will lead to a consistent trend of higher rankings for LLM-generated documents by PLM-based retrievers, further confirming the observations in Figure 1.

> **Finding 1:** For PLM-based retrievers, document perplexity has a causal effect on estimated relevance scores. Lower perplexity can lead to higher relevance scores.

## 4.2 ANALYZING MECHANISM BEHIND THE BIASED EFFECT

### 4.2.1 WHY PERPLEXITY AFFECTS PLM-BASED RETRIEVERS?

In Section 4.1, our empirical experiments have confirmed that PLM-based retrievers take perplexity features into account for document retrieval. However, the reason why perplexity-related features play a role, particularly when these models are primarily designed for document ranking, remains unclear. Considering that PLM-based retrievers are generally fine-tuned from PLMs on retrieval tasks, we delve into the relationship between the mask language modeling task in the pre-training stage and the mean-pooling document retrieval task in the fine-tuning stage. Our formulation are as follows and explanations can be found in Appendix C.1.

**Model Architecture.** To simplify our analysis, we assume a common architecture for PLM-based retrievers, consisting of an encoder $f(\boldsymbol{t}; \boldsymbol{\theta}) : \mathcal{T}^{L \times D} \mapsto \mathcal{R}^{L \times N}$ and a one-layer decoder $g(\boldsymbol{z}; \boldsymbol{W}) = \sigma(\boldsymbol{z}\boldsymbol{W})$, where $\mathcal{T}$ denotes the set composed of one-hot vectors, $L$ is the length of query or document, $D$ is the dictionary size, $N$ is the dimension of embedding vector, and $\sigma(\cdot)$ maps real vectors to simplexes. For the ease of qualitative analysis, we replace softmax operation with a linear operation, and $\boldsymbol{z}\boldsymbol{W}$ is assumed positive to ensure the well-definition of the probability distribution.

**Masked Language Modeling (MLM) Task.** The PLM is initially pre-trained on the MLM task with CrossEntropy loss: $\mathcal{L}_1(\boldsymbol{d}) = -\frac{1}{L}\mathbf{1}_L^T[\boldsymbol{d} \odot \log g(f(\boldsymbol{d}))]\mathbf{1}_D$, where $\odot$ denotes the Hadamard product, $\frac{1}{L}\mathbf{1}_L$ means averages over the length of the documents, $[\boldsymbol{d} \odot \log g(f(\boldsymbol{d}))]\mathbf{1}_D$ is the expression of CrossEntropy using one-hot vectors.

**Document Retrieval Task.** In the fine-tuning stage for the document retrieval task, the retrieval model estimates the relevance for given query-document pairs by computing the dot product of the document embedding vectors $\boldsymbol{d}^{\mathrm{emb}} = f(\boldsymbol{d}, \boldsymbol{\theta})$ and query embedding vectors $\boldsymbol{q}^{\mathrm{emb}} = f(\boldsymbol{q}, \boldsymbol{\theta})$. Without loss of generality, we assume $\|\boldsymbol{d}_l^{\mathrm{emb}}\|_2 = 1, \ l = 1, \ldots, L$, which means the embeddings of each token is normalized. The loss function can be written as $\mathcal{L}_2(\boldsymbol{d}, \boldsymbol{q}) = -tr[(\frac{1}{L}\mathbf{1}_L\boldsymbol{d}^{\mathrm{emb}})^T(\frac{1}{L}\mathbf{1}_L\boldsymbol{q}^{\mathrm{emb}})]$, where $\frac{1}{L}\mathbf{1}_L[\cdot]$ is the mean pooling operation of the embeddings over the document length $L$.

With the formulation above, we further explore the theoretical underpinnings of why perplexity influences retrieval performance by examining the gradients of the loss functions for both the MLM task and the document retrieval task, as shown in the following Theorem 1:

**Theorem 1.** *Given the following three conditions:*

• *Representation Collinearity: the embedding vectors of relevant query-document pairs are collinear after mean pooling, i.e.,*

$$\mathbf{1}_{L \times L} f(\boldsymbol{q}) = \lambda \mathbf{1}_{L \times L} f(\boldsymbol{d}), \lambda > 0.$$

• *Semi-Orthogonal Weight Matrix: the weight matrix of the decoder is semi-orthogonal, i.e.,*

$$\boldsymbol{W}\boldsymbol{W}^T = \boldsymbol{I}_N.$$

• *Encoder-decoder Cooperation: fine-tuning does not disrupt the corresponding function between encoder and decoder, i.e.,*

$$f(\boldsymbol{d}) = g^{-1}(\boldsymbol{d}).$$

*Then there exists a matrix* $\boldsymbol{K} = \left[\frac{\lambda k_l}{L(1-k_l)}\right]_{ln} \in \mathcal{R}_+^{L \times N}, k_l = \sum_d^D (\boldsymbol{d}^{\mathrm{emb}} \boldsymbol{W})_{ld}$ *which satisfies*

$$\frac{\partial \mathcal{L}_2}{\partial \boldsymbol{d}^{\mathrm{emb}}} = \boldsymbol{K} \odot \frac{\partial \mathcal{L}_1}{\partial \boldsymbol{d}^{\mathrm{emb}}}.$$

The three conditions made with their rationale are explained in Appendix C.1 and the proof of Theorem 1 can be found in Appendix C.2. From Theorem 1, we observe that the gradients of the two losses of MLM task and the retrieval task have a positive linear relationship.

Note that $\mathcal{L}_1(\boldsymbol{d})$ actually represent the document perplexity $P_d$ and $\mathcal{L}_2(\boldsymbol{d}, \boldsymbol{q})$ actually represent the negative estimated relevance score $-\hat{R}_{q,d}$. Then we can easily derive the following Corollary, which illustrates how the key conclusion $\partial \mathcal{L}_2 / \partial \boldsymbol{d}^{\mathrm{emb}} = \boldsymbol{K} \odot \partial \mathcal{L}_1 / \partial \boldsymbol{d}^{\mathrm{emb}}$ in Theorem 1 leads to the biased effect of document perplexity $P_d$ on estimated relevance score $\hat{R}_{q,d}$:

**Corollary 1.** *Consider a human-written document $\boldsymbol{d}_1$ and its LLM-rewritten document $\boldsymbol{d}_2$, they are both relevant with query $\boldsymbol{q}$. Assume LLM-rewritten documents possess lower perplexity at token level (Mitchell et al., 2023). Let* rvec/vec *be matrix-to-row/column-vector operator, $\mathcal{L}_1^l(\boldsymbol{d})$ denote the perplexity of the $l$-th token in the document, $(\boldsymbol{d}_2^{\mathrm{emb}})_l$ denote the embedding of the $l$-th token,*

$$\mathcal{L}_1^l(\boldsymbol{d}_1) - \mathcal{L}_1^l(\boldsymbol{d}_2) = \frac{\partial \mathcal{L}_1(\boldsymbol{d}_2)}{\partial (\boldsymbol{d}_2^{\mathrm{emb}})_l} \cdot \frac{\partial (\boldsymbol{d}_2^{\mathrm{emb}})_l}{\partial \boldsymbol{d}_2} \cdot \mathrm{vec}(\boldsymbol{d}_1 - \boldsymbol{d}_2) > 0, \;\; l = 1, \dots, L,$$

*where 1st-order approximation of Chain rule is taken as the surrogate function (Grabocka et al., 2019; Nguyen et al., 2009) for $\mathcal{L}_1^l(\boldsymbol{d})$. According to Theorem 1 and 1st-order approximation of $\mathcal{L}_2(\boldsymbol{d})$,*

$$\hat{R}_{q,d_1} - \hat{R}_{q,d_2} = -[\mathcal{L}_2(\boldsymbol{d}_1) - \mathcal{L}_2(\boldsymbol{d}_2)] = -\mathrm{rvec}(\boldsymbol{K} \odot \frac{\partial \mathcal{L}_1(\boldsymbol{d}_2^{\mathrm{emb}})}{\partial \boldsymbol{d}_2^{\mathrm{emb}}}) \cdot \frac{\partial \boldsymbol{d}_2^{\mathrm{emb}}}{\partial \boldsymbol{d}_2} \cdot \mathrm{vec}(\boldsymbol{d}_1 - \boldsymbol{d}_2)$$

$$= -\sum_{l=1}^{L} \frac{\lambda k_l}{L(1-k_l)} \frac{\partial \mathcal{L}_1(\boldsymbol{d}_2)}{\partial (\boldsymbol{d}_2^{\mathrm{emb}})_l} \frac{\partial (\boldsymbol{d}_2^{\mathrm{emb}})_l}{\partial \boldsymbol{d}_2} \mathrm{vec}(\boldsymbol{d}_1 - \boldsymbol{d}_2) = -\sum_{l=1}^{L} \frac{\lambda k_l}{L(1 - k_l)} \left(\mathcal{L}_1^l(\boldsymbol{d}_1) - \mathcal{L}_1^l(\boldsymbol{d}_2)\right) < 0.$$

Corollay 1 indicates that human-written document will receive lower relevance estimation than its LLM-written document, resulting in source bias. It is important to note that our theoretical analysis does not cover all situations in reality, we will discuss these limitations in Appendix B.

> **Finding 2:** For PLM-based retrievers, the gradients of MLM and IR loss functions (metrics) possess linear overlap, leading to the biased effect of perplexity on estimated relevance scores.

### 4.2.2 FURTHER VERIFICATION OF THEOREM 1

Theorem 1 reveals the linear relationship between language modeling gradients and retrieval gradients w.r.t. document embedding vectors $\boldsymbol{d}^{\mathrm{emb}}$. For a more comprehensive verification for its reliability, we derive Corollay 2 from Theorem 1 and provide supporting experiments about the corollary. The derivation is similar with that in Corollay 1.

**Corollary 2.** *For two retrievers $f(\boldsymbol{t}; \boldsymbol{\theta}_1), f(\boldsymbol{t}; \boldsymbol{\theta}_2)$ which share the same PLM, such as BERT. If retriever $f(\boldsymbol{t}; \boldsymbol{\theta}_1)$ possesses more powerful language modeling ability than $f(\boldsymbol{t}; \boldsymbol{\theta}_2)$, i.e.,*

$$\mathbb{E}_{\boldsymbol{d} \in \mathcal{D}}[\mathcal{L}_1^l(\boldsymbol{d}; \boldsymbol{\theta}_1)] - \mathbb{E}_{\boldsymbol{d} \in \mathcal{D}}[\mathcal{L}_1^l(\boldsymbol{d}; \boldsymbol{\theta}_2)] < 0, \;\; l = 1, \dots, L,$$

*then similar to the Corollary 1, we have*

$$\mathbb{E}_{\boldsymbol{d}\in\mathcal{D}}[\mathcal{L}_2(\boldsymbol{d};\boldsymbol{\theta}_1)] - \mathbb{E}_{\boldsymbol{d}\in\mathcal{D}}[\mathcal{L}_2(\boldsymbol{d};\boldsymbol{\theta}_2)] = \mathbb{E}_{\boldsymbol{d}\in\mathcal{D}}\left[\text{rvec}\left(\frac{\partial\mathcal{L}_2(\boldsymbol{d}^{\text{emb}};\boldsymbol{\theta}_2)}{\partial\boldsymbol{d}^{\text{emb}}}\right)\cdot\frac{\partial\boldsymbol{d}^{\text{emb}}}{\partial\boldsymbol{\theta}_2}\cdot\text{vec}(\boldsymbol{\theta}_1 - \boldsymbol{\theta}_2)\right]$$

$$=\mathbb{E}\left[\text{rvec}(\boldsymbol{K}\odot\frac{\partial\mathcal{L}_1(\boldsymbol{d}^{\text{emb}};\boldsymbol{\theta}_2)}{\partial\boldsymbol{d}^{\text{emb}}})\frac{\partial\boldsymbol{d}^{\text{emb}}}{\partial\boldsymbol{\theta}_2}\text{vec}(\boldsymbol{\theta}_1 - \boldsymbol{\theta}_2)\right] = \mathbb{E}\left[\sum_{l=1}^{L}\frac{\lambda k_l}{L(1-k_l)}\left(\mathcal{L}_1^l(\boldsymbol{d};\boldsymbol{\theta}_1) - \mathcal{L}_1^l(\boldsymbol{d};\boldsymbol{\theta}_2)\right)\right] < 0.$$

Note that $\mathbb{E}_{\boldsymbol{d}\in\mathcal{D}}[\mathcal{L}_1(\boldsymbol{d};\boldsymbol{\theta})]$ is a typical measure of language modeling ability and $\mathbb{E}_{\boldsymbol{d}\in\mathcal{D}}[\mathcal{L}_2(\boldsymbol{d};\boldsymbol{\theta})]$ reflects the ranking performance, Corollary 2 indicates that if a retriever possesses more powerful language modeling ability, its ranking performance will be better.

To offer empirical support for the corollary, we evaluate the language modeling ability of PLM-based retrieval models with different ranking performances. By taking the retrieval model directly as a PLM encoder to do MLM task, we calculate the average text perplexity of the retrieval corpus to evaluate their language modeling ability, which offers support for the encoder-decoder corporation assumption at the same time. As illustrated in Figure 3, there is a clear correlation between text perplexity and retrieval accuracy (except Contriever). These results, demonstrating that language modeling capabilities are indeed correlated with retrieval performance, strengthen the practical reliability of our assumptions and conclusions as the deductive verification of the above hypothesis we used. This finding also explains why PLM dramatically improve the performance of retrievers over past years.

Combining the previous findings, we can further understand the relationship between model retrieval performance and the degree of source bias. On one hand, if the PLM-based retriever demonstrates a better MLM capability, it tends to be more sensitive to document perplexity, which leads to more severe source bias (Corollary 1). On the other hand, a retriever with better MLM capabilities can also achieve more accurate relevance estimations, leading to better ranking performance (Corollary 2). Consequently, PLM-based retrievers encounter a trade-off between accuracy in retrieval and the severity of source bias. Specifically, higher ranking performance is associated with more significant source bias. This relationship has been noted in previous research (Dai et al., 2024a), and we are the first to offer a plausible explanation for this phenomenon.

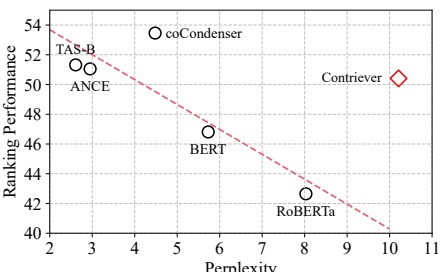

Figure 3: Model perplexity and ranking performance (NDCG@3) on averaged results of DL19, TREC-COVID, and SCIDOCS.

> **Finding 3:** Better language modeling improves PLM-based retriever's ranking performance, but also heightens its sensitivity to perplexity, thus increasing source bias severity.

## 5 CAUSAL-INSPIRED SOURCE BIAS MITIGATION

In this section, we further propose a causal-inspired debiasing method to eliminate the source bias, which can be naturally derived from our above causal analysis. We then conduct experiments to evaluate the effectiveness of the proposed debiasing method.

### 5.1 PROPOSED DEBIASING METHOD: CAUSAL DIAGNOSIS AND CORRECTION

In Section 3 and 4, we have constructed a causal graph and estimated the biased effect of perplexity on the final predicted relevance score. Based on these insights, we propose an inference-time debiased method via **C**ausal **D**iagnosis **C**orrection (CDC). The main procedure of CDC lies on two stage: (i) *Bias Diagnosis*: Employing the Instrumental Variable method for estimating the bias effect $\hat{\beta}_2$ of perplexity $P_d$ to estimated relevance score $\hat{R}_{q,d}$. (ii) *Bias Correction*: Separating the biased effect of document perplexity from the overall estimated relevance scores $\hat{R}_{q,d}$.

---

**Algorithm 1:** The Proposed CDC: Debiasing with Causal Diagnosis and Correction

---

**Input:** training set $\mathcal{D}$, test query set $\mathcal{Q}$, test corpus $\mathcal{C}$, estimation budget $M$
**Output:** unbiased estimated relevance scores $\tilde{\mathcal{R}}$

```
1  // Bias Diagnosis
```
2  Initialize the estimation set for estimating biased effect $\mathcal{D}_e \leftarrow \emptyset$
3  **for** *training pairs* $(q_i, d_i^{\mathcal{H}}) \in \mathcal{D}$ *and* $|\mathcal{D}_e| < M$ **do**
4      Instruct LLM to generate doc $d_i^{\mathcal{G}}$ via rewriting the original human-written doc $d_i^{\mathcal{H}}$
5      Predict the estimated relevance scores $\hat{r}_i^{\mathcal{H}}, \hat{r}_i^{\mathcal{G}}$ for pairs $(q_i, d_i^{\mathcal{H}})$ and $(q_i, d_i^{\mathcal{G}})$
6      Calculate perplexity $p_i^{\mathcal{H}}, p_i^{\mathcal{G}}$ for doc $d_i^{\mathcal{H}}$ and doc $d_i^{\mathcal{G}}$, respectively
7      Updating the estimation set $\mathcal{D}_e \leftarrow \mathcal{D}_e \cup (\hat{r}_i^{\mathcal{H}}, \hat{r}_i^{\mathcal{G}}, p_i^{\mathcal{H}}, p_i^{\mathcal{G}})$
8  **end**
9  Estimate the biased effect coefficient $\hat{\beta}_2$ with 2-stage regression using Eq. (2) on $\mathcal{D}_e$
```
10  // Bias Correction
```
11  **for** *test query* $q_t \in \mathcal{Q}$ **do**
12      Predict the estimated relevance scores $\hat{r}_t$ for each pair $(q_t, d_t)$ with $d_t \in \mathcal{C}$
13      Calculate document perplexity $p_t$ for each doc $d_t \in \mathcal{C}$
14      Debias the original model prediction $\hat{r}_t$ using Eq. (3), add the calibrated score $\tilde{r}_t$ to $\tilde{\mathcal{R}}$
15  **end**
16  **return** $\tilde{\mathcal{R}}$

---

Table 2: Performance (NDCG@3) and bias (Relative $\Delta$ (Dai et al., 2024c) on NDCG@3) of different PLM-based retrievers with and without our proposed CDC debiased method on three datasets. Note that a more negative bias metric value indicates a greater bias towards LLM-generated documents, while a more positive value indicates a greater bias towards human-written documents.

| Model | DL19 (In-Domain) | | | | TREC-COVID (Out-of-Domain) | | | | SCIDOCS (Out-of-Domain) | | | |
| | Performance | | Bias | | Performance | | Bias | | Performance | | Bias | |
| | Raw | +CDC | Raw | +CDC | Raw | +CDC | Raw | +CDC | Raw | +CDC | Raw | +CDC |
|---|---|---|---|---|---|---|---|---|---|---|---|---|
| BERT | 75.92 | 77.65 | -23.68 | 5.90 | 53.72 | 45.88 | -39.58 | -18.40 | 10.80 | 10.44 | -2.85 | 29.19 |
| Roberta | 72.79 | 71.33 | -36.32 | 4.45 | 46.31 | 45.86 | -48.14 | -10.51 | 8.85 | 8.24 | -30.90 | 32.13 |
| ANCE | 69.41 | 67.73 | -21.03 | 34.95 | 71.01 | 69.94 | -33.59 | -1.94 | 12.73 | 12.31 | -1.57 | 26.26 |
| TAS-B | 74.97 | 75.63 | -49.17 | -9.97 | 63.95 | 62.84 | -73.36 | -37.42 | 15.04 | 14.15 | -1.90 | 23.48 |
| Contriever | 72.61 | 73.83 | -21.93 | -5.33 | 63.17 | 61.35 | -62.26 | -31.33 | 15.45 | 15.09 | -6.96 | 1.63 |
| coCondenser | 75.50 | 75.36 | -18.99 | 9.60 | 70.94 | 71.07 | -67.95 | -45.39 | 13.93 | 13.79 | -5.95 | 1.06 |

Specifically, the final calibrated score $\tilde{R}_{q,d}$ for document ranking can be formulated as follows:

$$\tilde{R}_{q,d} = \hat{R}_{q,d} - \hat{\beta}_2 P_d, \tag{3}$$

which can be derived by rearranging Eq. (2). In this formula, $\tilde{R}_{q,d}$ is independent to document source and perplexity. Therefore, it serves as a good proxy for semantic relevance ranking.

Specifically, we first take $M$ samples from the training set $\mathcal{D}$ to construct the estimation set $\mathcal{D}_e$ for estimating the biased effect $\hat{\beta}_2$ (lines 2-8), where $M$ is the estimation budget. To construct the estimation set $\mathcal{D}_e$, we instruct an LLM to generate document $d_i^{\mathcal{G}}$ by rewriting the original human-written document $d_i^{\mathcal{H}}$. For these two types of samples, we use the retriever to predict their relevance scores $\hat{r}_i^{\mathcal{H}}$ and $\hat{r}_i^{\mathcal{G}}$ for the given query and calculate their document perplexities $p_i^{\mathcal{H}}$ and $p_i^{\mathcal{G}}$. Further, following the practice in Section 4.1, we use two-stage IV regression on the estimation set $\mathcal{D}_e$ to estimate the biased coefficient $\hat{\beta}_2$ (line 9). During testing, we use Eq. (3) to correct the original model prediction $\hat{r}_t$, obtaining the calibrated score $\tilde{r}_t$ for final document ranking (line 11-15). We summarize the overall procedure of CDC in Algorithm 1.

## 5.2 EXPERIMENTS AND ANALYSIS

To evaluate the effectiveness of CDC, we implement it across various retrievers in simulated-realistic debiasing scenarios, where generated documents are from different domains and LLMs. In this case, we investigate the generalizability of the CDC method at both LLM level and Domain level.

At domain-level, we employ bias diagnosis on the training set of DL19 to estimate the biased effect $\hat{\beta}_2$ for each retrieval model, and then conduct in-domain and cross-domain evaluation on the test sets

Table 3: Performance (NDCG@3) and bias (Relative $\Delta$ (Dai et al., 2024c) on NDCG@3) of the retrievers on mixed SciFact corpus from different LLMs. Bias Diagnosis is conducted on DL19 corpus from Llama-2, where CDC performs generalization at both LLM and data-domain levels.

| Model | Llama-2 (In-Domain) | | | | GPT-4 (Out-of-Domain) | | | | GPT-3.5 (Out-of-Domain) | | | | Mistral (Out-of-Domain) | | | |
| | Performance | | Bias | | Performance | | Bias | | Performance | | Bias | | Performance | | Bias | |
| | Raw | +CDC | Raw | +CDC | Raw | +CDC | Raw | +CDC | Raw | +CDC | Raw | +CDC | Raw | +CDC | Raw | +CDC |
|---|---|---|---|---|---|---|---|---|---|---|---|---|---|---|---|---|
| BERT | 35.67 | 35.08 | -12.37 | 6.75 | 36.47 | 35.75 | -3.69 | 6.04 | 35.97 | 35.27 | -5.03 | 18.08 | 35.13 | 35.08 | 0.73 | 13.07 |
| RoBERTa | 38.09 | 36.76 | -29.54 | -0.88 | 38.53 | 37.70 | -11.98 | 4.52 | 39.17 | 38.00 | -35.39 | 14.09 | 38.29 | 37.28 | -17.95 | 16.78 |
| ANCE | 42.13 | 42.13 | -8.81 | 4.59 | 42.67 | 42.99 | -5.53 | 3.28 | 42.76 | 42.96 | -13.59 | 6.09 | 42.62 | 42.71 | -8.59 | 1.82 |
| TAS-B | 52.95 | 53.94 | -15.04 | -7.96 | 52.12 | 52.44 | -4.94 | -0.05 | 52.83 | 52.90 | -5.65 | 5.57 | 52.18 | 52.69 | -8.71 | -2.00 |
| Contriever | 55.19 | 55.37 | -2.87 | 1.07 | 55.78 | 55.70 | -5.32 | -4.44 | 56.11 | 56.17 | -7.43 | -2.81 | 56.13 | 56.28 | -4.13 | -2.39 |
| coCondenser | 49.53 | 49.40 | -12.98 | -9.26 | 48.57 | 48.91 | 5.04 | 6.04 | 48.59 | 48.81 | -1.00 | 5.30 | 49.57 | 49.92 | -5.90 | -0.76 |

of DL19, TREC-COVID, SCIDOCS. Note that only 128 samples (i.e., estimation budget $M = 128$) are used for bias diagnosis, this sample size is sufficient for effective results. More detailed settings can be found in Appendix E.1. The averaged results over five different seeds are reported in Table 2.

As we can see, using the estimated biased coefficient of in-domain retrieval data, our debiasing CDC successfully mitigates or even reverses the retrieval models' preference towards human-written documents without fine-tuning retrievers. Meanwhile, this estimated biased coefficient demonstrates generalizability across out-of-domain datasets. The majority of the retrieval performance degradation was generally less than 2 percentage points, revealing that our debiasing CDC has acceptable impact on ranking performance, see detailed significance test in Appendix Table 7. In addition, the mean and standard deviation of performance and bias after CDC debiasing for the five sampling sessions is provided in Appendix Table 6, indicating the robustness of CDC to training samples.

We also find that the debiasing results may vary across different retrievers. Specifically, CDC has more significant effects on vanilla models like BERT while exhibits lower impacts on stronger retrievers such as Contriever. We offer the following analysis to explain such observations. Stronger retrievers are developed using more sophisticated contrastive learning algorithms, which enhance their abilities to differentiate between highly relevant documents and the others. In this way, it may be more challenging for CDC corrections to alter the initial rankings. So a more aligned or model-specific approach could potentially enhance the debiasing process.

Considering that the web content may be generated by diverse LLMs, we expand our evaluations to assess the generalizability of the CDC method across corpora generated by different LLMs, including Llama (Touvron et al., 2023), GPT-4 (Achiam et al., 2023), GPT-3.5, and Mistral (Jiang et al., 2023). Due to the cost of computing resources, we conducted experiments on a smaller dataset SciFact, which is also used in previous works (Dai et al., 2024c;a). In this setup, CDC used Llama's rewritten DL19 documents to estimate $\beta_2$ and subsequently correct retrieval results on SciFact corpora mixed with each LLM separately. The results displayed in Table 3 confirm that CDC is capable to generalize across various LLMs and maintain high retrieval performance while effectively mitigating bias.

In summary, these empirical results validate the feasibility of our proposed debiasing method by effectively reducing the biased impact of document perplexity on model outputs. And this method can be integrated efficiently into dual-encoder architerctures used in ANN search by pre-computing and indexing query-independent document perplexity with embeddings. Moreover, $\hat{\beta}_2$ can be adjusted according to specific requirements, where a larger absolute value of $\hat{\beta}_2$ leads to further preference for human-written texts albeit at the potential cost of ranking performance degradation. More discussion about the open question that "Should we debias toward human-written contents?" is in Appendix A.1.

## 6 CONCLUSION

This paper aims to explain the phenomenon of source bias where PLM-based retrievers overrate low-perplexity documents. Our core conclusion is that PLM-based retrievers use perplexity features for relevance estimation, leading to source bias. To verify this, we conducted a two-stage IV regression and found a negative causal effect from perplexity to relevance estimation. Theoretic analysis reveals that the gradient correlation between language modeling and retrieval tasks contributes to this causal effect. Based on the analysis, a causal-inspired inference-time debiasing method called CDC is proposed. Experimental results verified its effectiveness in terms of debiasing the source bias.

## 7 ACKNOWLEDGEMENTS

This work was funded by the National Key R&D Program of China (2023YFA1008704), the National Natural Science Foundation of China (62472426, 62276248, 62376275), the Youth Innovation Promotion Association CAS under Grants (2023111), fund for building world-class universities (disciplines) of Renmin University of China, the Fundamental Research Funds for the Central Universities, PCC@RUC, and the Research Funds of Renmin University of China (RUC24QSDL013). Work partially done at Engineering Research Center of Next-Generation Intelligent Search and Recommendation, Ministry of Education.

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

# A   DISCUSSION

## A.1   SHOULD WE DEBIAS TOWARD HUMAN-WRITTEN CONTENTS?

While we refer to the retrievers' preference for LLM-rewritten content as a "bias", it's crucial to recognize that not all biases are harmful. As illustrated in previous works (Dai et al., 2024c; Zhou et al., 2024), from content creators' perspective, reducing preference toward LLM-rewritten content helps guarantee sufficient incentives for authors to encourage creativity, and thus sustain a healthier content ecosystem. From users' perspective, LLM-rewritten documents might possess enhanced quality, such as better coherence, and improved reading experience.

In this work, our debiasing approach is primarily a methodological application derived from our causal graph analysis, serving to validate the "perplexity-trap" hypothesis further. At the same time, our framework allows for adjustable preference levels between human-written and LLM-generated documents, catering to specific practical requirements. This flexibility ensures our approach can be tailored to balance between enhancing information quality and maintaining content provider fairness.

## A.2   SHOULD PERPLEXITY BE A CAUSAL FACTOR TO QUERY-DOCUMENT RELEVANCE?

It's one of the assumptions of this work that perplexity should not be a causal factor to query-document relevance. It is true that there may be a correlation between perplexity and query-document relevance, e.g., the coherence of a document may also have an impact on relevance. However, there is an insurmountable gap between the perplexity of LLM-rewritten documents and human work, because people do not intentionally take PPL into account when writing, but LLMs do generation with perplexity as a goal. We are currently faced with a situation where this perplexity gap has breached the range of human perception of relevance, leading to serious source bias even when the rewritten documents share nearly the same semantics with human works, as verified and discussed in previous literature (Dai et al., 2024c). It's just like what Goodhart's Law (Goodhart, 1975) states: "When a measure becomes a target, it ceases to be a good measure." So perhaps a threshold should be set, and when perplexity is less than the threshold, it should be made independent with relevance.

# B   LIMITATIONS

This study has several limitations that are important to acknowledge.

**Data and Experiments.**     Firstly, while our analysis was conducted on three representative datasets, it is recognized that there are numerous other IR datasets that could have been included. Our selection, although limited in scope, was strategic to ensure a broad representation across different domains, and we believe that our findings can be generalized to other domains. Secondly, due to the cost associated with human evaluation, we were constrained to perform only $6 \times 20$ evaluations for each dataset, corresponding to six different sampling temperatures. This decision, while pragmatic, may limit the extent to which we can generalize our results to other conditions. Thirdly, we have to admit that impacts of LLM rewriting on semantics indeed lack more consideration although they have been designed possibly credible. Since simulated environment construction is not our main contribution, we have adopted the datasets provided by previous works (Dai et al., 2024b) or follow their methodology (Dai et al., 2024c) to evaluate the source bias of retrievers. In Dai et al. (2024c), the document embeddings are compared using cosine similarity, and a more detailed human evaluation was conducted to assess the various impacts of LLM rewriting, which indicated no significant changes in document semantics. We will conduct more meticulous semantic checks to pursue more rigorous conclusions if possible.

**Theoretical Analysis**     In our theoretical proofs, we made certain assumptions and simplifications. Specifically, we narrow our analysis in PLM-based dual-encoder and mean-pooling scenario. These are necessary to achieve mathematical tractability and are grounded in practical considerations, which have been discussed in the previous sections. We believe these assumptions are reasonable and have validated the reliability of our conclusions through experimental verification. For the other scenarios, such as auto-regressive embedding models and CLS-based retrievers, we will explore and discuss them in the future work.

Despite these limitations, we maintain that our work provides valuable insights into the subject matter and serves as a foundation for future research.

## C    NOTES ON THEORETICAL ANALYSIS

In this section, we provide detailed reasons to our assumptions and proof to our proposed theorem in Section 4.2.

### C.1    EXPLANATION ON ASSUMPTIONS

Our theoretical analysis are based on a set of assumptions, to which we're going to offer the reasons.
• **Encoder-Only Retrievers**     Encoder-only architectures are generally considered more suitable for textual representation tasks, while encoder-decoder and decoder-only models are typically used for generative tasks. Thus, encoder-only models have been widely employed for retrieval tasks and have demonstrated effective results. In fact, most of the mainstream dense retrievers listed on the MTEB (Muennighoff et al., 2022) leaderboard are based on encoder-only architectures.
• **Mean-Pooling Strategy.**     We use of mean pooling for query/doc embeddings in the derivation of Theorem 1, while a simplification, differs from the practice of using CLS token embeddings in BERT-like models. From a practical perspective, (weighted) mean pooling embedding outperform CLS token embedding when ranking, which has been widely confirmed in previous works (Dai et al., 2024b; Reimers, 2019). From a theoretical perspective, (weighted) mean pooling is able to retain more local information about documents, which is important for retrieval tasks, as a query is regarded related to a document when the query is related to a particular sentence in the document. Furthermore, there is literature indicating that CLS token embeddings may not always effectively capture sentence representations, which can be a limitation in retrieval contexts (Li et al., 2020).
• **Representation Collinearity Hypothesis**     Representation Collinearity Hypothesis is a fundamental assumption long implemented in information retrieval systems (Salton et al., 1975). When measuring relevance scores by calculating dot or cosine similarity, we assume that the best relevant document owns an embedding that is collinear with the query embedding (given that the norm of the document embedding is held constant). In practice, dense retrievers are trained on contrastive learning to maximize the similarity between query and its relevant documents while minimize the similarity of irrelevant documents (Gao et al., 2021; Zhao et al., 2024).
• **Semi-Orthogonal Weight Matrix Hypothesis**     $W \in \mathbb{R}^{N \times D}$ satisfies the Semi-Orthogonal Weight Matrix assumption $\boldsymbol{W}\boldsymbol{W}^T = \boldsymbol{I}_N$, which is necessary to achieve mathematical tractability. Since practical PLMs uses 2-layer MLPs rather than the weight matrix $W$, this can't be verified directly. If we ignore the activation function in the MLPs of BERT, let $\boldsymbol{W} = \boldsymbol{W}_1\boldsymbol{W}_2$, then $\frac{1}{N^2}\|\boldsymbol{W}\boldsymbol{W}^T\|_F \approx 50 \cdot \frac{1}{N^2}\|\text{diag}(\boldsymbol{W}\boldsymbol{W}^T)\|_F$, which suggests that the diagonal elements are much larger than the others. One reasonable intuition is a conclusion in high-dimension probabilities which states "for any $\epsilon > 0$, there are $m = \Omega(e^N)$ vectors in $\mathcal{R}^N$ such that any pair of them are nearly orthogonal."(Mitzenmacher and Upfal, 2017) Since $N \geq 768$ for commonly-used retrievers, the hypothesis holds with high probability.
• **Encoder-decoder Cooperation Hypothesis**     This assumption has a certain practical background. The experiment in Section4 can be viewed as a verification of this assumption, where finetuned encoder is used with unfinetuned MLPs to do MLM task. In this setting, the hybrid model recieve relatively low perplexity as PLMs. In practice, the beginning learning rate of finetuning retrievers is usually set at $1 \cdot e^{-5}$, which makes retrievers more likely to the conserve of inversion property.

### C.2    PROOF OF THEOREM 1

In this section, we provide the proof of Theorem 1. Note that the three conditions made are naturally satisfied: (1) Representation collinearity is a fundamental assumption long implemented in information retrieval systems. (2) Matrix orthogonality is a common and intuitive property of the decoder's weight matrix. (3) Encoder-decoder adheres to the original design principles of auto-encoder networks. Then we give the proof as follows:

*Proof.* Given the following three conditions:

- Representation Collinearity: the embedding vectors of relevant query-document pairs are collinear after mean pooling, i.e.,

$$\mathbf{1}_{L \times L} f(\boldsymbol{q}) = \lambda \mathbf{1}_{L \times L} f(\boldsymbol{d}).$$

- Orthogonal Weight Matrix: the weight matrix of the decoder is orthogonal, i.e.,

$$\boldsymbol{W} \boldsymbol{W}^T = \boldsymbol{I}.$$

- Encoder-decoder cooperation: fine-tuning does not disrupt the corresponding function between encoder and decoder, i.e.,

$$f(\boldsymbol{d}) = g^{-1}(\boldsymbol{d}).$$

Our goal is to prove $\partial \mathcal{L}_2 / \partial \boldsymbol{d}^{\mathrm{emb}} = \boldsymbol{K} \odot \partial \mathcal{L}_1 / \partial \boldsymbol{d}^{\mathrm{emb}}$.

Note that the two losses are both involved with $\boldsymbol{d}^{\mathrm{emb}}$,

$$\frac{\partial \mathcal{L}_1}{\partial \boldsymbol{d}^{\mathrm{emb}}} = -\frac{1}{L} \mathbf{1}_{L \times L} [g(\boldsymbol{d}^{\mathrm{emb}}) - \boldsymbol{d}] \boldsymbol{W}^T = -\frac{1}{L} \mathbf{1}_{L \times L} [\sigma(\boldsymbol{d}^{\mathrm{emb}} \boldsymbol{W}) - \boldsymbol{d}] \boldsymbol{W}^T.$$

$$\frac{\partial \mathcal{L}_2}{\partial \boldsymbol{d}^{\mathrm{emb}}} = -\frac{1}{L^2} \mathbf{1}_{L \times L} \boldsymbol{q}^{\mathrm{emb}}.$$

Replacing the softmax operation with linear normalization, let $\div$ denote element-wise division,

$$[\sigma(\boldsymbol{x})]_l = \frac{1}{\sum_n^N \boldsymbol{x}_{ln}} \boldsymbol{x}_l, \quad l = 1, \dots, L.$$

Considering the following matrix identity,

$$(A_{M \times N} \cdot B_{N \times K}) \odot (\boldsymbol{c}_M \cdot \mathbf{1}_K^T) = (A_{M \times N} \odot (\boldsymbol{c}_M \cdot \mathbf{1}_N^T)) \cdot B_{N \times K},$$

it reveals that the gradient of $\mathcal{L}_1$ can be rearranged as

$$[\sigma(\boldsymbol{d}^{\mathrm{emb}} \boldsymbol{W}) - \boldsymbol{d}] \boldsymbol{W}^T = (\frac{\boldsymbol{d}^{\mathrm{emb}} \boldsymbol{W}}{\boldsymbol{k}_L \mathbf{1}_D^T} - \boldsymbol{d}) \boldsymbol{W}^T = (\frac{\boldsymbol{d}^{\mathrm{emb}}}{\boldsymbol{k}_L \mathbf{1}_N^T} \boldsymbol{W} - \boldsymbol{d}) \boldsymbol{W}^T,$$

where column vector $\boldsymbol{k}_L \in \mathbb{R}^L$ satisfies $k_l = \sum_d^D (\boldsymbol{d}^{\mathrm{emb}} \boldsymbol{W})_{ld} > 0$ because $\sigma(\boldsymbol{d}^{\mathrm{emb}} \boldsymbol{W})_l$ is a complex. Meanwhile, using mean inequality (also called QM-AM inequality), we can find that

$$k_l \leq \sqrt{\frac{1}{N} \sum_d^D (\boldsymbol{d}^{\mathrm{emb}} \boldsymbol{W})_{ld}^2} = \sqrt{\frac{1}{N} (\boldsymbol{d}^{\mathrm{emb}} \boldsymbol{W})_l (\boldsymbol{d}^{\mathrm{emb}} \boldsymbol{W})_l^T} = \frac{1}{\sqrt{N}} \|\boldsymbol{d}_l^{\mathrm{emb}}\|_2 = \frac{1}{\sqrt{N}} < 1.$$

According to the orthogonal weight matrix assumption,

$$(\frac{\boldsymbol{d}^{\mathrm{emb}}}{\boldsymbol{k}_L \mathbf{1}_N^T} \boldsymbol{W} - \boldsymbol{d}) \boldsymbol{W}^T = \frac{\boldsymbol{d}^{\mathrm{emb}}}{\boldsymbol{k}_L \mathbf{1}_N^T} - \boldsymbol{d} \boldsymbol{W}^T = \frac{\boldsymbol{d}^{\mathrm{emb}}}{\boldsymbol{k}_L \mathbf{1}_N^T} - \sigma^{-1}(\boldsymbol{d}) \boldsymbol{W}^T = \frac{\boldsymbol{d}^{\mathrm{emb}}}{\boldsymbol{k}_L \mathbf{1}_N^T} - g^{-1}(\boldsymbol{d}).$$

From the encoder-decoder cooperation condition, we obtain

$$\frac{\partial \mathcal{L}_1}{\partial \boldsymbol{d}^{\mathrm{emb}}} = -\frac{1}{L} \mathbf{1}_{L \times L} [\frac{\boldsymbol{d}^{\mathrm{emb}}}{\boldsymbol{k}_L \mathbf{1}_N^T} - f(\boldsymbol{d})] = -\frac{1}{L} \mathbf{1}_{L \times L} [\boldsymbol{d}^{\mathrm{emb}} \odot (\frac{\mathbf{1}_L - \boldsymbol{k}_L}{\boldsymbol{k}_L} \mathbf{1}_N^T)] = -\frac{1}{L} \mathbf{1}_{L \times L} \mathrm{diag}(\frac{\mathbf{1}_L - \boldsymbol{k}_L}{\boldsymbol{k}_L}) \boldsymbol{d}^{\mathrm{emb}}.$$

Considering the positive query-document pair $q, d$, assume their embedding vectors are collinear,

$$\frac{\partial \mathcal{L}_2}{\partial \boldsymbol{d}^{\mathrm{emb}}} = -\frac{1}{L^2} \mathbf{1}_{L \times L} \boldsymbol{q}^{\mathrm{emb}} = -\frac{\lambda}{L^2} \mathbf{1}_{L \times L} \boldsymbol{d}^{\mathrm{emb}}.$$

we can observe that

$$\frac{\partial \mathcal{L}_2}{\partial \boldsymbol{d}^{\mathrm{emb}}} = \frac{\lambda}{L} \mathrm{diag}(\frac{\boldsymbol{k}_L}{\mathbf{1}_L - \boldsymbol{k}_L}) \frac{\partial \mathcal{L}_1}{\partial \boldsymbol{d}^{\mathrm{emb}}}.$$

Let $\boldsymbol{K} = \frac{\lambda}{L} \frac{\boldsymbol{k}_L}{\mathbf{1}_L - \boldsymbol{k}_L} \mathbf{1}_N^T$, then it holds that

$$\frac{\partial \mathcal{L}_2}{\partial \boldsymbol{d}^{\mathrm{emb}}} = \boldsymbol{K} \odot \frac{\partial \mathcal{L}_1}{\partial \boldsymbol{d}^{\mathrm{emb}}}.$$

$\square$

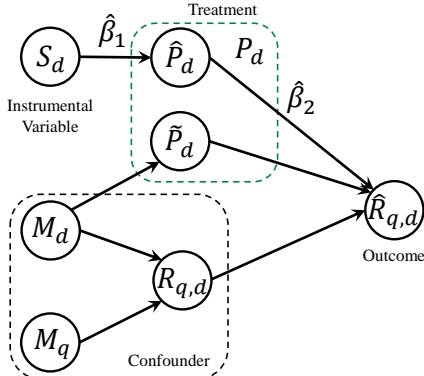

Figure 4: By leveraging IV regression on $S_d$, $P_d$ is decomposed into causal and non-causal parts. A precise causal effect can be obtained from the coefficient of the second-stage regression, i.e., $\hat{\beta}_2$.

## D    Instrumental Variable Regression

In statistics, instrumental variable (IV) is used to estimate causal effects. The changes of IV induces changes of explanatory variable but keeps error term constant. The basic method to estimate causal effect is 2SLS. In the first stage, 2SLS regress explanatory variable on instrumental variable and obtain the predicted values of explanatory variable. In the second stage, 2SLS regress output variable on predicted explanatory variable. Then, the coefficient corresponding to the predicted explanatory variable can be viewed as a measure of causal effects.

According to our proposed causal graph, the document source $S_d$ has three properties: (1) It is correlated to the document perplexity $P_d$. (2) It is independent with Document semantics $M_d$ because we can instruct LLMs to rewrite human documents for any document semantics. (3) It only affect the estimated relevance score $\hat{R}_{q,d}$ through document perplexity $P_d$. Thus, document source $S_d$ can be considered a instrumental variable to evaluate the causal effect of document perplexity on estimated relevance scores.

As depicted in Figure 4, we estimate document perplexity $P_d$ based on document source $S_d$ in the first stage. The results, coefficient $\hat{\beta}_1$ and predicted document perplexity $\hat{P}_d = \hat{\beta}_1 S_d$, are used in the second stage to estimated the predicted relevance score $\hat{R}_{q,d}$ via linear regression, where the estimated coefficient $\hat{\beta}_2$ is a valid measure for the magnitude of the causal effect.

## E    More Experiments

### E.1    Experimental Details

Our experiments are all conducted on machines equipped with NVIDIA A6000 GPUs and 52-core Intel(R) Xeon(R) Gold 6230R CPUs at 2.10GHz. For better reproducibility, we employ the following officially released checkpoints:

BERT (Devlin et al., 2018; 2019) and RoBERTa (Liu et al., 2019) are used in dense retrieval as PLM encoders. We employ the trained models from the Cocktail benchmark (Dai et al., 2024a). The models are available at `https://huggingface.co/IR-Cocktail/bert-base-uncased-mean-v3-msmarco` and `https://huggingface.co/IR-Cocktail/roberta-base-mean-v3-msmarco`, respectively.

ANCE (Xiong et al., 2020) improves dense retrieval by sampling hard negatives via the Approximate Nearest Neighbor (ANN) index. The model is available at `https://huggingface.co/sentence-transformers/msmarco-roberta-base-ance-firstp`.

TAS-B (Hofstätter et al., 2021), leverages balanced margin sampling for efficient query selection. The model is available at `https://huggingface.co/sentence-transformers/msmarco-distilbert-base-tas-b`.

Table 4: Human evaluation on which document is more relevant to the given query semantically? The numbers in parentheses are the proportion agreed upon by all three human annotators.

| Temperature | DL19 | | |
| --- | --- | --- | --- |
| | Human | LLM | Equal |
| 0.00 | 0.0% (0.0%) | 5% (0.0%) | 95% (83.8%) |
| 0.20 | 0.0%(0.0%) | 5% (0.0%) | 95% (94.2%) |
| 0.40 | 0.0% (0.0%) | 0.0% (0.0%) | 100% (79.6%) |
| 0.60 | 0.0% (0.0%) | 0.0% (0.0%) | 100% (84.6%) |
| 0.80 | 0.0% (0.0%) | 0.0% (0.0%) | 100% (94.5%) |
| 1.00 | 0.0%(0.0%) | 0.0% (0.0%) | 100% (94.5%) |
| Temperature | TREC-COVID | | |
| | Human | LLM | Equal |
| 0.00 | 0.0% (0.0%) | 0.0% (0.0%) | 100% (84.6%) |
| 0.20 | 0.0% (0.0%) | 0.0% (0.0%) | 100% (94.5%) |
| 0.40 | 0.0% (0.0%) | 0.0% (0.0%) | 100% (74.6%) |
| 0.60 | 0.0% (0.0%) | 0.0% (0.0%) | 100% (94.5%) |
| 0.80 | 0.0% (0.0%) | 0.0% (0.0%) | 100% (79.6%) |
| 1.00 | 0.0% (0.0%) | 0.0% (0.0%) | 100% (84.6%) |
| Temperature | SCIDOCS | | |
| | Human | LLM | Equal |
| 0.00 | 0.0% (0.0%) | 0.0% (0.0%) | 100% (84.6%) |
| 0.20 | 0.0% (0.0%) | 0.0% (0.0%) | 100% (84.6%) |
| 0.40 | 0.0% (0.0%) | 0.0% (0.0%) | 100% (79.6%) |
| 0.60 | 0.0% (0.0%) | 5.0% (0.0%) | 95% (83.8%) |
| 0.80 | 0.0% (0.0%) | 0.0% (0.0%) | 100% (79.6%) |
| 1.00 | 0.0% (0.0%) | 5% (0.0%) | 95% (89.0%) |

Contriever (Izacard et al., 2022) employs contrastive learning with positive samples generated through cropping and token sampling. The model is available at `https://huggingface.co/facebook/contriever-msmarco`.

coCondenser (Gao and Callan, 2022) is a retriever that conducts both pre-training and supervised fine-tuning. The model is available at `https://huggingface.co/sentence-transformers/msmarco-bert-co-condensor`.

We follow the metrics proposed by previous work when measuring ranking performance and source bias. For ranking performance, we use NDCG@$k$ (Järvelin and Kekäläinen, 2002). For source bias, we use Relative $\Delta$ NDCG@$k$ (Dai et al., 2024a;c; Xu et al., 2024; Zhou et al., 2024), which is formulated as

$$Relative\Delta = \frac{Metric_{Human} - Metric_{LLM}}{\frac{1}{2}(Metric_{Human} + Metric_{LLM})} \times 100\%.$$

In CDC debiasing, considering the sample size we conduct bias correction for the top 10 candidates in retrieval. Rising up the candidates number leads to less preference for LLM-generated documents while ranking performance may drop a little.

### E.2 MORE RESULTS OF GENERATED CORPUS WITH VARYING SAMPLING TEMPERATURE

#### E.2.1 HUMAN EVALUATION

Although LLM-generated documents are solely based on their corresponding human documents, it is still necessary to verify that the generated document has the same relevance scores with given query as the original documents. To provide empirical support on the fact that LLM-generated documents are not injected with extraneous information about queries, we conduct a human evaluation.

We randomly select 20 (query, human-written document, LLM-generated document) triples for each dataset and each sampling temperature. The human annotators who have at least Bachelor's degrees are asked to evaluate which document is more relevant without knowing document sources. Their results are transferred into the "Human", "LLM", or "Equal" options later. The final labels of each

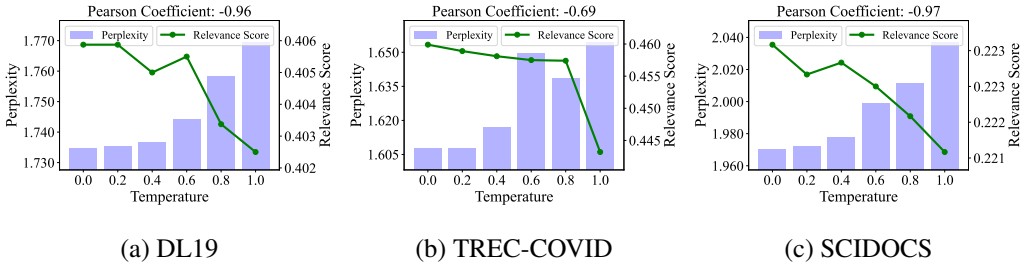

Figure 5: Perplexity and estimated relevance scores of Contriever on positive query-document pairs in three datasets, where documents are generated by LLM with different sampling temperatures.

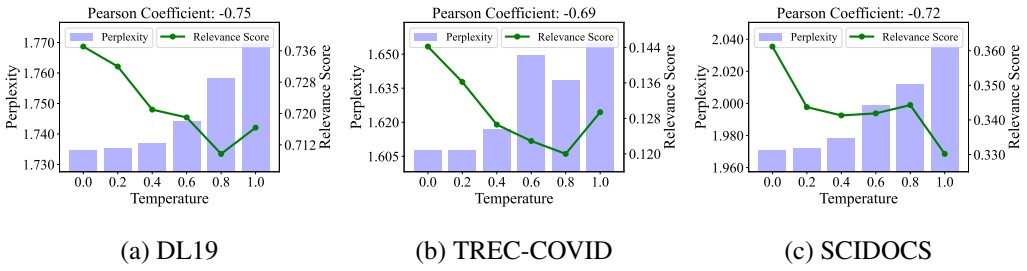

Figure 6: Perplexity and estimated relevance scores of TAS-B on positive query-document pairs in three datasets, where documents are generated by LLM with different sampling temperatures.

triple are determined by the votes of three different annotators. The results in Table 4 illustrate that documents from different sources possess the same relevance to the corresponding queries, which guarantees the correctness of our controlled variables experiments.

### E.2.2 RESULTS WITH MORE PLM-BASED RETRIEVERS

In Section 3.1 we discover the negative correlation between document perplexity and estimated relevance scores by Contriever. In this section, we demonstrate the replicability of the discovery by providing similar results on TAS-B and Contriever. As depicted in Figure 5 and Figure 6, there is a significant negative correlation between document perplexity and estimated relevance scores as sampling temperature changes. Documents with lower perplexity obtain prevalent higher estimated relevance scores across different PLM-based retrievers, further affirming the universality of the phenomenon.

### E.2.3 MORE RESULTS OF $\beta_2$ ESTIMATION

In Section 4.1, we estimated the causal effect of perplexity on estimated relevance scores through 2SLS. Since the estimation needs LLM generation, it's natural to explore the hyperparameters related to the generation.

According to the causal graph we proposed, the sampling temperature does affect $\hat{\beta}_1$ in the first stage of the regression, but is independent with $\hat{\beta}_2$. We explore whether $\hat{\beta}_1$ changes in turn affects the value of $\hat{\beta}_2$ by using documents generated with different sampling temperature. The $\hat{\beta}_1$ and $\hat{\beta}_2$ obtained from our estimation on the set of rewritten texts with different sampling temperatures are shown in Table 5. It can be found that under the maximum sampling temperature difference, the variation of $\hat{\beta}_1$ is within 15% and the variation of $\hat{\beta}_2$ is within 20%, and such variations are similar to the errors brought by random sampling, so the variation of the sampling temperature is acceptable in the CDC algorithm.

Table 5: The influence of generation temperatures on the magnitude of the causal coefficients $\beta_1, \beta_2$. The coefficients are estimated from all positive query-document pairs.

| | DL19 | | | | TREC-COVID | | | | SCIDOCS | | | |
|---|---|---|---|---|---|---|---|---|---|---|---|---|
| Temperature | 0.0 | 0.2 | 0.4 | 0.6 | 0.0 | 0.2 | 0.4 | 0.6 | 0.0 | 0.2 | 0.4 | 0.6 |
| $\hat{\beta}_2$(BERT) | -7.80 | -7.78 | -7.77 | -7.94 | -1.21 | -1.20 | -1.24 | -1.26 | -2.29 | -2.29 | -2.33 | -2.46 |
| $\hat{\beta}_2$(RoBERTa) | -23.57 | -23.50 | -23.45 | -23.97 | 1.73 | 1.73 | 1.77 | 1.80 | -6.02 | -6.04 | -6.13 | -6.47 |
| $\hat{\beta}_2$(ANCE) | -0.44 | -0.44 | -0.44 | -0.45 | 0.07 | 0.07 | 0.07 | 0.07 | -0.22 | -0.22 | -0.22 | -0.23 |
| $\hat{\beta}_2$(TAS-B) | -0.81 | -0.80 | -0.80 | -0.82 | -0.34 | -0.34 | -0.35 | -0.35 | -0.37 | -0.37 | -0.37 | -0.39 |
| $\hat{\beta}_2$(Contriever) | -0.01 | -0.01 | -0.01 | -0.01 | -0.03 | -0.03 | -0.04 | -0.04 | -0.02 | -0.02 | -0.02 | -0.02 |
| $\hat{\beta}_2$(coCondenser) | -0.58 | -0.58 | -0.58 | -0.59 | -0.23 | -0.23 | -0.24 | -0.24 | -0.25 | -0.25 | -0.25 | -0.26 |
| $\hat{\beta}_1$ | -0.44 | -0.44 | -0.44 | -0.43 | -0.41 | -0.41 | -0.40 | -0.39 | -0.41 | -0.40 | -0.40 | -0.38 |

## E.3    MORE RESULTS OF CDC DEBIASING

In this section, we report more experimental results to provide a more comprehensive analysis of CDC, including robustness analysis with error bar (Table 6) and significance test (Tabel 7).

Table 6: Mean and standard deviation of Performance (NDCG@3) and bias (Relative $\Delta$ (Dai et al., 2024c) on NDCG@3) of different PLM-based retrievers with our proposed CDC debiased method on three datasets in five repetitions.

| Model | DL19 (In-Domain) | | | | TREC-COVID (Out-of-Domain) | | | | SCIDOCS (Out-of-Domain) | | | |
|---|---|---|---|---|---|---|---|---|---|---|---|---|
| | Performance | | Bias | | Performance | | Bias | | Performance | | Bias | |
| | Mean | Std | Mean | Std | Mean | Std | Mean | Std | Mean | Std | Mean | Std |
| BERT | 77.65 | 0.89 | 5.90 | 4.40 | 45.88 | 1.14 | -18.40 | 6.72 | 10.44 | 0.19 | 29.19 | 9.35 |
| RoBERTa | 71.33 | 0.48 | 4.45 | 0.80 | 45.86 | 0.78 | -10.51 | 3.58 | 8.24 | 0.18 | 32.13 | 7.28 |
| ANCE | 67.73 | 0.15 | 34.95 | 11.51 | 69.94 | 0.77 | -1.94 | 4.63 | 12.31 | 0.33 | 26.26 | 10.61 |
| TAS-B | 75.63 | 0.24 | -9.97 | 5.25 | 62.84 | 0.48 | -37.42 | 3.99 | 14.15 | 0.16 | 23.48 | 5.84 |
| Contriever | 73.83 | 0.27 | -5.33 | 1.93 | 61.35 | 0.73 | -31.33 | 3.22 | 15.09 | 0.10 | 1.63 | 1.89 |
| coCondenser | 75.36 | 0.47 | 9.60 | 8.49 | 71.07 | 0.45 | -45.39 | 8.55 | 13.79 | 0.21 | 1.06 | 2.79 |

Table 7: The $p$-value of significance test conducted on the NDCG@3 and Relative $\Delta$ (Dai et al., 2024c) on NDCG@3 with and without CDC debias method, with bold fonts indicating the Performance or Bias can pass a significance test with $p$-value$< 0.05$. As expected, most Performance DOES NOT pass the significance test while all the Bias DOES pass the significance test.

| Model | DL19 | | TREC-COVID | | SCIDOCS | |
|---|---|---|---|---|---|---|
| | Performance | Bias | Performance | Bias | Performance | Bias |
| BERT | **4.56e-03** | **5.33e-04** | **1.87e-03** | **3.97e-05** | 1.37e-01 | **1.47e-03** |
| RoBERTa | 7.26e-02 | **2.33e-04** | 1.22e-01 | **5.46e-05** | 8.48e-01 | **9.45e-07** |
| ANCE | 1.74e-01 | **4.48e-03** | 4.61e-01 | **3.09e-04** | 1.62e-01 | **2.25e-05** |
| TAS-B | 6.16e-01 | **3.19e-04** | 8.58e-01 | **1.27e-03** | **6.67e-04** | **2.16e-04** |
| Contriever | 2.77e-01 | **3.94e-02** | 2.98e-01 | **5.03e-04** | **4.44e-02** | **7.65e-03** |
| coCondenser | 8.82e-01 | **1.16e-02** | 5.95e-01 | **3.81e-04** | 2.71e-01 | **1.58e-03** |

