# OpenReview forum: "Perplexity Trap: PLM-Based Retrievers Overrate Low Perplexity Documents"
_ICLR.cc/2025/Conference — ICLR 2025 Poster_

### Official Review · Reviewer_S3rs · 2024-10-27

**Soundness:** 3
**Presentation:** 3
**Contribution:** 2
**Rating:** 5
**Confidence:** 4

**Summary:**

The paper studies why PLM-based retrievers are biased towards LLM-generated content and hypothesize that it is due to a causal effect between document perplexity and relevance score estimation. Based on the analysis, the paper further proposes a correction method, CDC, to separate the bias effect. Experiments include controlled experiments to demonstrate the causal effect, and experiments on 3 IR datasets, using existing MLM-based PLM retrievers, where the effect of using CDC is demonstrated.

**Strengths:**

1. Performing a deeper understanding on the PLM retriever is biased towards AIGC is an interesting topic and can be interesting to some from the IR community. The paper provides reasonable illustrations via causal analysis and proposes a debiasing method accordingly, which is a complete work in terms of the story.

2. The analysis of MLM vs IR loss function is interesting and provides more insights into the problem.

**Weaknesses:**

1. Limited to PLM-based retrievers. One may argue that a well behaved retriever, or existing retrievers do not possess this issue. Or there're more standard ways to address this such as by standard training. One may argue that this topic is not of general interest to the ICLR community as it is more SIGIR focused, as the setting is specific and not widely acknowledged.

2. Analysis is limited to MLM, while MLM is becoming less and less relevant. Indeed, the retrievers used in the paper are relatively old (most recent 2022). The proposed method also shows more effect on older PLMs than more recent ones. This again raises the concern about how relevant the paper is to the broader ML community. How about models trained with CLM?

3. The causal analysis and approach leveraged in the paper is conventional. This is not necessarily a "weakness" but it does not make the paper a stronger case.

**Questions:**

Would not "document semantic" be explained away?

Please list the statistics of each dataset used.

---

> ### Author Response · Authors · 2024-11-18
> **Response to Reviewer S3rs : Part 1**
>
> Thanks for acknowledging our work and the constructive comments. Please kindly find point-to-point responses below.
> > W1: Limited to PLM-based retrievers. One may argue that a well behaved retriever, or existing retrievers do not possess this issue. Or there're more standard ways to address this such as by standard training. One may argue that this topic is not of general interest to the ICLR community as it is more SIGIR focused, as the setting is specific and not widely acknowledged.
>
> **Response:** Thanks for the comments from the ML perspective. Below are our point-by-point clarification:
> 1. Previous studies[1, 2] have found that neural retrievers prevalently exhibit bias towards LLM-generated documents. One contribution of this paper lies in providing a more rigorous experimental validation and theoretical analysis of such empirical conclusions, narrowing the statements that PLM-based retrievers exhibit bias towards documents with low PPL, thus offering a clearer definition of the source bias. Except frequency-based models perfering human-written documents, it remains unclear whether non-pretrained models exhibit similar biases, which is a question for future research.
> 2. There may be numerous approaches to address this issue such as adding a debiased training constraint [1]. We presents a tractable method based on our causal analysis, its effectiveness  strengthens our explanations. We believe that these explanations can also inspire model training techniques such as synthesizing training data and designing training objectives,  leading to emergement of more elegant solutions in the future.
> 3. In this paper, we verify that the text embeddings (representations) produced by fine-tuned PLMs may have inherent biases in downstream retrieval task, further explanation on which inductive bias leads to this phenomenon are provided in our theoretical analysis. Thus, we believe this analysis aligns with the theme of ICLR. Historically, many papers featured in causal retrieval or recommendation[3, 4, 5] are accepted in ICLR, so we believe our topic is suitable to ICLR based on past outcomes.
>
> > W2: Analysis is limited to MLM, while MLM is becoming less and less relevant. Indeed, the retrievers used in the paper are relatively old (most recent 2022). The proposed method also shows more effect on older PLMs than more recent ones. This again raises the concern about how relevant the paper is to the broader ML community. How about models trained with CLM?
>
> **Response:** Thanks for the inspiring comments.  we will explain this from two aspects.
> 1. We indeed initially focused on analysis on MLMs for the following reasons: BERT-style MLM architectures are generally considered more suitable and widely used in textual representation tasks such as infomation retrieval, as we have discussed in the Appendix C1. In fact, latest SOTA retrievers from MTEB benchmark are MLM-based [6, 7].
> 2. Following your feedback, we have expanded our analysis to include other architectures like CLM and find that our theoretical framework remains applicable. We find that although different architectures utilize diverse contextual information in token prediction, this variation does not alter the fundamental methods for calculating perplexity and relevance scores. More specifically, in Corollary 1, the variation between architectures only affects the matrix of partial derivatives of the embedding vectors with respect to the input text $\frac{\partial \boldsymbol{d}_2^{\mathrm{emb}}}{\partial \boldsymbol{d}_2}$, but does not impact the first-order gradient term of the loss function with respect to the embedding vectors $\frac{\partial \mathcal{L}}{\partial \boldsymbol{d}^{\mathrm{emb}}}$. Therefore, Corollary 1 still holds.
>
> We will include these expanded findings and present a more universally applicable conclusion in the final version. Thanks for your valuable comments again.
>
> **For more responses to the Weaknesses and Questions, please see the next part response.** We hope the above response can address your concerns and look forward to further discussions. Thanks again!
>
> [1] Neural Retrievers are Biased Towards LLM-Generated Content, KDD2024
>
> [2] Cocktail: A Comprehensive Information Retrieval Benchmark with LLM-Generated Documents Integration, ACL2024
>
> [3] Causal Fairness under Unobserved Confounding: A Neural Sensitivity Framework, ICLR2024
>
> [4] Calibration Matters: Tackling Maximization Bias in Large-scale Advertising Recommendation Systems, ICLR2023
>
> [5] From Intervention to Domain Transportation: A Novel Perspective to Optimize Recommendation, ICLR2022
>
> [6] jina-embeddings-v3: Multilingual Embeddings With Task LoRA, arXiv: 2409.10173
>
> [7] mGTE: Generalized Long-Context Text Representation and Reranking Models for Multilingual Text Retrieval, arXiv: 2407.19669

---

> ### Author Response · Authors · 2024-11-18
> **Response to Reviewer S3rs : Part 2**
>
> > W3: The causal analysis and approach leveraged in the paper is conventional. This is not necessarily a "weakness" but it does not make the paper a stronger case.
>
> **Response:** Thanks for the discussion about the causal analysis approach we used. 2SLS based on instrumental variables is a conventional yet well-developed and widely-used tool for causal inference. The application of this approach to the novel domain of source bias supports the credible verification of our conclusion in the paper, also demonstrating the effectiveness of this statistical method. Appendix D provides a comprehensive exposition on how this method fits our analysis of source bias, indicating our conclusion is appropriate and persuasive.
>
> > Q1: Would not "document semantic" be explained away?
>
> **Response:** Thanks for the question on our causal graph-based explanation, where document semantic is assumed to indirectly affect estimated relevance scores. Some  explanation on rationales behind each edge in the causal graph lies in Section 3.2. We will explain further analysis from two aspects.
> 1. In Section 3.1, we discuss that LLM-rewritten documents possess almost identical semantic as their origin. This is supported by the human evaluations in Appendix E2.1 and embedding similarity comparasion presented in [1]. Therefore, source bias cannot be directly attributed to document semantics alone.
> 2. Meanwhile, to mitigate the impact of subtle semantic differences, we conducted a causal analysis based on 2SLS, a method that is able to tolerate changes in document semantics. So considering changes in document semantics does not affect the reliability of our conclusions.
>
> > Q2: Please list the statistics of each dataset used.
>
> **Response:** We provide the detailed statistics of experimental datasets in the follow table:
> | Dataset   | Domain         | Task                  | Relevancy | Query Num. | Document Num.  |
> |-----------|----------------|-----------------------|-----------|-------------|---------------|
> | DL19      | Misc.          | Passage-Retrieval     | Binary    | 43          | 542,203       |
> | TREC-COVID| Bio-Medical    | Bio-Medical IR         | 3-level   | 50          | 128,585       |
> | SCIDOCS   | Scientific     | Citation-Prediction   | Binary    | 1,000       | 25,259        |
>
> **Thanks again for the comments and questions. We hope the response can address your concerns and look forward to further discussions.**
>
> [1] Neural Retrievers are Biased Towards LLM-Generated Content, KDD2024

---

> ### Author Response · Authors · 2024-11-27
>
> Dear Reviewer,
>
> First of all, we would like to sincerely thank you for taking the time to review our paper and provide valuable feedback. Your comments have been incredibly helpful in improving the content of the paper.
> As we are currently in the rebuttal stage, we would kindly like to remind you that if you have any further suggestions or feedback on our response, we would greatly appreciate it if you could share them by the 27th. After this date, we will no longer be able to make modifications to the PDF based on the reviewers' comments. Your continued guidance is crucial for us to refine the paper.
> Once again, thank you for your hard work and support, and we look forward to your valuable response.
>
> Best regards,
>
> Authors

---

> ### Comment · Reviewer_S3rs · 2024-11-29
>
> Thanks for the reply. After reviewing the reply and other reviews, I decide to keep the score. I'm still not fully convinced about the timeliness, broad interest, and significance of this work.

---

> > ### Author Response · Authors · 2024-11-30
> >
> > Dear Reviewer,
> >
> > Thanks for your response.  We would make another attempt once more to clarify the timeliness, broad interest, and significance of this work.
> >
> > 1. For the significance of this work, we focus on an emerging but underestimated bias in this LLM era and try to mitigate the unfair problem it causes, which is also acknowledged by other reviewers. In this LLM era, such bias is a significant damage to human contributors' utility and may leads to their exit from content creation platforms [1-2].
> > 2. Though source bias in previous works have received much attention, we are the first to provide both experimental and theoretical explanation to its existance [3-5]. Our findings enlighten researchers about the inductive bias of PLM-based representation learning and inference, offering potential direction for model training and deployment in the future.
> > 3. As for the timelineness, the models and methodology are representive and suitable, as we have pointed out that PLM-based retrievers are still dominant and widely used in practical retrieval systems nowadays [6].
> >
> > Besides the general explanation above, we're aslo looking forward to concrete questions to make the discussion more effective. So please don't hesitate to tell us if any problem remains to help us improve the quality of this paper. Thanks again for your reply.
> >
> > [1] Human vs. Generative AI in Content Creation Competition: Symbiosis or Conflict? ICML2024
> >
> > [2] How to Strategize Human Content Creation in the Era of GenAI? 2024
> >
> > [3] Neural Retrievers are Biased Towards LLM-Generated Content, KDD2024
> >
> > [4] Spiral of Silences: How is Large Language Model Killing Information Retrieval? -- A Case Study on Open Domain Question Answering. ACL 2024
> >
> > [5] Invisible Relevance Bias: Text-Image Retrieval Models Prefer AI-Generated Images. SIGIR2024
> >
> > [6] Dense Text Retrieval based on Pretrained Language Models: A Survey. TOIS 2024

---

### Official Review · Reviewer_kkqd · 2024-10-29

**Soundness:** 3
**Presentation:** 4
**Contribution:** 4
**Rating:** 6
**Confidence:** 3

**Summary:**

In this article, the authors propose a thorough study of the "Perplexity Trap," a phenomenon previously observed: when using embeddings from pretrained language models for retrieval tasks, documents written by LLMs tend to be ranked higher. The main hypothesis is that this occurs because generated texts tend to have lower perplexity due to the generative process, which in turn affects document/query similarity. The authors aim to confirm this hypothesis through three experimental studies (one providing a contribution to mitigate this bias) and one theoretical study. All results seem to confirm the perplexity hypothesis.

**Strengths:**

Very important topic to study, including a consensual hypothesis for the “perplexity trap” that was still underexplored

Both experimental (correlation 3.1 and causality 4.1) and theoretical findings are (quite) convincing explanation of the link between perplexity and IR document relevance

The proposed method for bias mitigation is relevant, new and effectively reduce bias on several dataset and LLMs.

**Weaknesses:**

My main concern is the theoretical development in section 4.2. It lacks formalism, some notations are badly introduced or not at all. This leads to some aspect (either details or important things) either difficult to understand and follow, or that seem false. I’ll developpe everything in the questions section below.

The rewriting approach appears to be sound but lacks experimental support to show that the semantic content was not changed. See a more detailed comment below.

**Questions:**

## On the rewriting method
“Specifically, we manipulate the sampling temperatures during generation to obtain LLM-generated documents with different PPLs but similar semantic content.” and “Since document semantics remain unchanged during rewriting,”.
-> This is a strong assumption, that is, in my opinion, not verified. Using high temperature could lead to texts with different semantic content. The article lacks an experimental verification using LLM based eval, human eval, LSA/LDA based approaches or Non LLM embedding similarity. The question asked to the human evaluator in E.2.1 only evaluate relevance (author should provide the complete question asked to the annotator) and not semantic equivalence, and evaluates only 20 triplets. Additionally, increasing temperature might lead to hallucination, mechanically decreasing the relevance of the document. These should be evaluated properly. Could you elaborate on this?

## On the theoretical section 4.2 and appendix

Notations should be formally introduced to ease the readability, such as \textbf{d} or dimensions of matrices (e.g. W). As notations are not that consensual, this is important to ease the reader work. Such grey zones led to several questions/interrogation that I list here.

Line 312, d_{emb} is an L \times N matrix, and is assumed to have a l2 norm of 1. What does it mean here for a matrix? Are you constraining the spectral norm, which is the formal equivalent of L2 norm for matrices? Could you please clarify.

Following the explanations, W is the matrix that maps a vector in \mathbb{R}^n to the vocabulary space, so of dim N by D.
1) It cannot be orthogonal but only semi orthogonal as it is not a squared matrix
2) I don’t understand how the conditions can be verified for most PLM as imposing orthogonality is really hard. Could you clarify? I might have missed something in the formalization of the problem, but it seems to be nowhere explained and motivated.

Last assumption (fine tuning conserve the inversion property) is not discussed and motivated. With large learning rate, I don’t see how this can be verified.

Line 810 is losing the reader. You are summing over N, if d^{emb}W is of dimension L \times D, so k is a vector of dimension L ? More importantly, I don’t get the justification for it having only positive values. If dW contains non-positive values, linear normalization does not provide guarantee that k is positive and that norm of d is 1 (again, which norm?). Although it relies on undisclosed assumptions that should be made clearer.

QM-AM states that the quadratic mean is greater than the arithmetic mean. K is n times the arithmetic mean, so you can’t say that this is lower than the quadratic mean (it is lower than  n times the QM though).

Overall, I find it misleading to use k, as it is a constant that does not depend from d_emb, that may lead to thinking that k is fixed for all d_emb.

Eventually, the sign of k will modify the dependency type and could contradict the assumption of the work.

Again, all these might come from a misunderstanding of the nature of matrix W, but anyway, this should be clarified in the documents.

## Minor

ANCE is used before being introduced line 264 (e.g. line 145 )
Shouldn’t MLM task loss divided by L only rather than L \times D ?

---

> ### Author Response · Authors · 2024-11-18
> **Response to Reviewer kkqd : Part 1**
>
> Thanks for acknowledging our work and the constructive comments. Please kindly find point-to-point responses below.
> > W2: The rewriting approach appears to be sound but lacks experimental support to show that the semantic content was not changed.
> > Q1:“Specifically, we manipulate the sampling temperatures during generation to obtain LLM-generated documents with different PPLs but similar semantic content.” and “Since document semantics remain unchanged during rewriting,”. -> This is a strong assumption, that is, in my opinion, not verified. Using high temperature could lead to texts with different semantic content. The article lacks an experimental verification using LLM based eval, human eval, LSA/LDA based approaches or Non LLM embedding similarity. The question asked to the human evaluator in E.2.1 only evaluate relevance (author should provide the complete question asked to the annotator) and not semantic equivalence, and evaluates only 20 triplets. Additionally, increasing temperature might lead to hallucination, mechanically decreasing the relevance of the document. These should be evaluated properly. Could you elaborate on this?
>
> **Response:** Thanks for the careful concern on the data. In our opinion, the impacts of LLM rewriting on semantics indeed lack more consideration although they have been designed possibly credible. Since simulated environment construction is not our main contribution, we have adopted the datasets provided by previous works [1, 2] or follow their methodology to evaluate the source bias of retrievers. In [1], the document embeddings  are compared using cosine similarity, and a more detailed human evaluation was conducted to assess the various impacts of LLM rewriting, which indicated no significant changes in document semantics. We will conduct more meticulous semantic checks to pursue more rigorous conclusions in the future.
>
> At the same time, the instrumental variables method we employed eliminates the interference of document semantic changes on the estimation of causal effects. In our causal graph explanation framework, document semantics is a confounder, whose changes do not affect the causal effect estimation of perplexity on the estimated relevant scores. Therefore, in spite of the changes of the documents semantics, the conclusions of perplexity trap remain credible.
>
> > W1: My main concern is the theoretical development in section 4.2. It lacks formalism, some notations are badly introduced or not at all. This leads to some aspect (either details or important things) either difficult to understand and follow, or that seem false.
>
> **Response:** Thanks for the comment on our theoretical analysis. Please allow us to briefly introduce our analytical approach once again. First, in Theorem 1, we found that if several specific conditions are met, there is a special linear relationship between the gradients of the MLM loss function and the IR loss with respect to document embeddings. Then, in Corollary 1, we found that if LLM-generated documents have lower perplexity at the token level, their estimated relevance score will be higher than that of their corresponding human-written counterparts, thus producing Source Bias. To illustrate the practicality and rationality of Theorem 1, we obtained Corollary 2 using a similar method, which indicates that the stronger a model's MLM capability, the stronger its IR capability. The phenomenon found in previous paper [1] and our empirical experiments both validate the conclusion in Corollary 2, further proving the rationality of the hypothesis we proposed. We have revised the analysis in the latest version of PDF with clearer expressions and we explain the notations with more details in the response to Q2.
>
> > Q3: ANCE is used before being introduced line 264 (e.g. line 145 ) Shouldn’t MLM task loss divided by L only rather than L \times D ?
>
> **Response:** Thanks for the careful notice. We will adjust the order in the final version to avoid declarations used before they are made. Additionally, the MLM loss  in the experimental section is divided by $L$  only. As for the theoretical analysis, we standardize it to be divided only by $L$ in the updated PDF, which doesn't affect the conclusions.
>
> **For more responses to the Weaknesses and Questions, please see the next comment.** We hope the above response can address your concerns and look forward to further discussions. Thanks again!
>
> [1] Neural Retrievers are Biased Towards LLM-Generated Content, KDD2024
>
> [2] Cocktail: A Comprehensive Information Retrieval Benchmark with LLM-Generated Documents Integration, ACL2024
>
> [3] Probability and computing: Randomization and probabilistic techniques in algorithms and data analysis, Cambridge university press2017.

---

> ### Author Response · Authors · 2024-11-18
> **Response to Reviewer kkqd : Part 2**
>
> > Q2: detailed question on (1) d_{emb} assumed to have a L2 norm. (2) weight matrix W is orthogonal assumption. (3) fine tuning conserve the inversion property assumption. (4) the explanation of k = sum_i(d^{emb}W)_i and related QA-AM.
>
> **Response:** Thanks for the patient concern on our theoretical analysis again. These questions is due to our imprecise use of notations. Here, we select some key points to provide detailed explanation, the rest of the content has also been revised line by line in the latest version of PDF with clearer expressions.
> 1. $\boldsymbol{d}^{\mathrm{emb}}\in\mathcal{R}^{L\times N}$. The L2 norm of $d^{emb}$ is actually $\|(\boldsymbol{d}^{emb})_l\|=1, \dots, L$, which means the embedding of each token is normalized.
> 2. $W \in \mathbb{R}^{N\times D}$ satisfies the Semi-Orthogonal Weight Matrix assumption $\boldsymbol{WW}^T = \boldsymbol{I}_N$, which is necessary to achieve mathematical tractability. Since practical PLMs uses 2-layer MLPs rather than the weight matrix $W$, this can't be verified directly. If we ignore the activation function in the MLPs of BERT, let $\boldsymbol{W}=\boldsymbol{W}_1\boldsymbol{W}_2$，then $\frac{1}{N^2}\|\boldsymbol{WW}^T\|_F \approx 50\cdot\frac{1}{N^2}\|\mathrm{diag}(\boldsymbol{WW}^T)\|_F$,  which suggests that the diagonal elements are much larger than the others. One reasonable intuition is a conclusion in high-dimension probabilities which states "for any $\epsilon>0$, there are $m=\Omega(e^N)$ vectors in $\mathcal{R}^N$ such that any pair of them are nearly orthogonal."[3] This part is now added in the Appendix C.1. Since $N \geq 768$ for commonly-used retrievers, the hypothesis holds with high probability.
> 3. Encoder-decoder cooperation assumption has a certain practical background. The experiment in Section 4.2.2 can be viewed as a verification of this assumption, where finetuned encoder is used with unfinetuned MLPs to do MLM task. In this setting, the hybrid model recieve relatively low perplexity as PLMs. In practice, the beginning learning rate of finetuning retrievers is usually set at $1\cdot e^{-5}$, which makes retrievers more likely to the conserve of inversion property. This part is aslo added in the Appendix C.1.
> 4. $\boldsymbol{d}\_{L}\in\mathbb{R}^L$ satisfies $k_l = \sum_d^D (\boldsymbol{d}^{\mathrm{emb}} \boldsymbol{W})_{ld}$ we claim $k_l > 0$ as an assumption in the latest version.
> 5. QM-AM states quadratic mean(the L2 norm of $(\boldsymbol{d}^\mathrm{emb}W)\_l$) is greater than the arithmetic mean ($k_{l} = \sum_d^D (\boldsymbol{d}^{\mathrm{emb}}\boldsymbol{W})_{ld}$).
>
> Hope our explanations are helpful in addressing your concerns . We sincerely apologize for any non-standard use of notations that may have caused confusion, and further in-depth discussions are welcomed to make theoretical analysis more clear.

---

> > ### Comment · Reviewer_kkqd · 2024-11-25
> > **Response to Authors**
> >
> > Dear authors,
> > Thank you for your careful response. I have read the paper revision and believe it helps. Nevertheless, I think you should take your time to motivate every hypothesis and make it clear in the paper.
> > I am willing to augment my score though.
> > Best regards,

---

> > > ### Author Response · Authors · 2024-11-25
> > >
> > > We are glad to know that the revision help make the analysis more convincing. In the revised paper, We have incoporated more explanation about each assumptions' rationale in Appendix C.1 and have detailed all assumptions in the Section 4.2.1 and Section 4.2.2. We have also clarified that the experiment in Section 4.2.2 can be viewed as a deductive verification of the assumption we used. Your enlightening comments and questions are sincerely appreciated, which provided constructive advice to make analysis more clear. Thanks again for your kindly support and further feedback. **However, we notice that the Rating seems to be unadjusted...... If there is any other concerns left, we are willing to try to clarify them as soon as possible.**

---

> ### Comment · Reviewer_kkqd · 2024-11-27
>
> Dear authors,
> I increased soundness contribution and score by 1
> Regards,

---

> > ### Author Response · Authors · 2024-11-27
> > **Thanks for the Response and Positive Feedback!**
> >
> > Thank you very much for your further positive feedback! We greatly appreciate the time and effort you dedicated to the review! Your valuable and insightful comments are really helpful in guiding us to improve our work.

---

### Official Review · Reviewer_V53D · 2024-11-04

**Soundness:** 3
**Presentation:** 3
**Contribution:** 4
**Rating:** 8
**Confidence:** 4

**Summary:**

This paper studies the causes for source bias in PLM-retrievers. It explores the underlying architectural relations between perplexity and relevance scores and proposes an inference-time mitigation method that decouples the perplexity bias from the retrieval task.

**Strengths:**

- A clear technical description of the source bias issue, experimentation for cause and architectural relation, clear presentation of results and limitations.
- The linear model fit for perplexity given document and semantic variables strengthens the argument presented by the causal graph.
- Human annotation for verifying synthetic correctness.
- I appreciate the acknowledgment and inclusion of semantic nuance of documents as another variable in your experiments.
- The offering of a mitigation method with CDC strengthens the argument presented in your study and is an appreciated contribution.

**Weaknesses:**

- The focus on small and older models like BERT and RoBerta is slightly exclusive of more recent LLMs and SLMs.
- Similarly, there is some lack of cross-model experiments for models used to generate data and for the retrieval task, popular closed- and open-source models could demonstrate very different results (A more inclusive experimental setup would be more convincing that your argument generalizes and isn’t overfit to your experiments).
- There might be a limitation with closed models where training data is unknown for the CDC methodology.

**Questions:**

- How relevant do you think your study will remain given the rise of synthetically trained models, given the effects of low ppl and high relevance on LLM-generated data by majorly human-data trained retrieval models in your experiments?

---

> ### Author Response · Authors · 2024-11-18
> **Response to Reviewer V53D :**
>
> Thanks for acknowledging our work and the constructive comments. Please kindly find point-to-point responses below.
>
> > W1: The focus on small and older models like BERT and RoBerta is slightly exclusive of more recent LLMs and SLMs.
>
> **Response:** Thanks for the inspiring comments.  we will explain this from two aspects.
> (1) We indeed initially focused on analysis on MLMs for the following reasons: BERT-style MLM architectures are generally considered more suitable and widely used in textual representation tasks such as infomation retrieval, as we have discussed in the Appendix C1. In fact, latest SOTA retrievers from MTEB benchmark are MLM-based [1, 2].
> (2) Following your feedback, we have expanded our analysis to include other architectures like CLM and find that our theoretical framework remains applicable. We find that although different architectures utilize diverse contextual information in token prediction, this variation does not alter the fundamental methods for calculating perplexity and relevance scores. More specifically, in Corollary 1, the variation between architectures only affects the matrix of partial derivatives of the embedding vectors with respect to the input text $\frac{\partial \boldsymbol{d}_2^{\mathrm{emb}}}{\partial \boldsymbol{d}_2}$, but does not impact the first-order gradient term of the loss function with respect to the embedding vectors $\frac{\partial \mathcal{L}}{\partial \boldsymbol{d}^{\mathrm{emb}}}$. Therefore, Corollary 1 still holds.
> We will include these expanded findings and present a more universally applicable conclusion in the final version. Thanks for your valuable comments again.
>
> > W2: Similarly, there is some lack of cross-model experiments for models used to generate data and for the retrieval task, popular closed- and open-source models could demonstrate very different results (A more inclusive experimental setup would be more convincing that your argument generalizes and isn’t overfit to your experiments).
>
> **Response:** Thanks for the advice. Considering that the web content maybe generated by diverse LLMs, we expand our evaluations to assess the generalizability of the CDC method across datasets generated by different LLMs,including Llama2, GPT-4, GPT-3.5, and Mistral. The results displayed in Table 3 in the PDF confirm that CDC is capable to generalize across various LLMs and maintain high retrieval performance while effectively mitigating bias. We will place greater emphasis on the cross-model generalization capabilities of CDC in the final version.
>
> > W3: There might be a limitation with closed models where training data is unknown for the CDC methodology.
>
> **Response:** Thanks for the insightful comments. Given that CDC method can generalize across different domains and different LLM-generated dataset, it is feasible to create training pairs on any human retrieval  datasets through LLM rewriting. Usually, 128 training instances are sufficient to effectively mitigate source bias, making the cost of constructing the training data acceptable. We will provide more details and explanation on constructing the training data for CDC in the final version.
>
> > Q1: How relevant do you think your study will remain given the rise of synthetically trained models, given the effects of low ppl and high relevance on LLM-generated data by majorly human-data trained retrieval models in your experiments?
>
> **Response:** Thanks for the enlightening question. Currently, there is a lack of in-depth exploration regarding the impact of training data on Source Bias in PLM-based retrievers. An intuitive analysis suggests that the Perplexity Trap is not caused by differences in PPL between positive and negative samples in the training data. This is because a document may be relevant(positive) to a query and irrelevant(negative) to another. The rise of synthetic training data may offer the following insights. On one hand, the main conclusions drawn so far are empirical findings from collected data, hence there are potential variables that may affect Source Bias. Using synthetic data to conduct controlled experiments is a promising direction for exploration,  of which our experiments in Section 3.1 can be viewed as a practice. On the other hand, our research can provide more inspiration for the training of retrieval models, such as how to synthesize training data to correct potential bias or train unbias model from scratch, similar to the use of counterfactual samples to enhance LLM training.
>
> We hope the above response can address your concerns and look forward to further discussions. Thanks again!
>
> [1] jina-embeddings-v3: Multilingual Embeddings With Task LoRA, arXiv: 2409.10173
>
> [2] mGTE: Generalized Long-Context Text Representation and Reranking Models for Multilingual Text Retrieval, arXiv: 2407.19669

---

> > ### Comment · Reviewer_V53D · 2024-11-26
> >
> > Thank you for your comprehensive and thoughtful responses. I appreciate your time and effort in making modifications to address them, completing the picture for me, especially your elaboration on cross-model experiments. Well done, overall, I will maintain my rating.

---

> > > ### Author Response · Authors · 2024-11-27
> > > **Thanks for the Response and Positive Feedback!**
> > >
> > > Thank you very much for your invaluable support and further positive feedback! We are glad to know that our responses and additional experiments have effectively addressed your concerns. We sincerely appreciate your constructive comments and questions, which help improve the quality of our paper.

---

### Author Response · Authors · 2024-11-24
**Summary of Rebuttal**

Dear Reviewers and ACs,

We sincerely thank all Reviewers and ACs for their great effort and constructive comments on our submission. During the discussion period, we have been focusing on these beneficial suggestions from the reviewers and doing our best to add several explanation and repsonse to reviewers. We believe our current response can help address all the reviewers' concerns.

As **all the reviewers** highlighted, our proposed causal graph-based explanation and our conclusion of perplexity trap is clear, reasonable, and reliable, which is supported by convincing experiments and enlightening theoretical analysis. Our proposed CDC debiasing method to address source bias is also appreciated by **all the reviewers**. We also appreciate that the reviewers acknowledge that our paper concentrates on an emerging but underestimated bias which is important in AIGC and IR  community(**Reviewer kkqd**, **Reviewer V53D**).

Moreover, we also thank the reviewers for pointing out the concerns regarding the scalability of our conclusion (**Reviewer V53D**, **Reviewer S3rs**), as well as the suggestions for expanding our discussion for the desirable properties of CDC(**Reviewer V53D**), the contribution of our work (**Reviewer S3rs**),  the rationale behind the subjects and methodologies employed in this paper (**Reviewer kkqd**, **Reviewer S3rs**), and more in-depth discussion about both the assumption and the notation in the theoretical analysis(**Reviewer kkqd**). We have carefully response to these constructive comments with the additional discussion:

● [Reviewer V53D, Reviewer S3rs] We have expanded our theoretical analysis to  retrievers and added some discussion about CLS token embedding in the limitation.

● [Reviewer V53D] We have implemented further explanation on the desirable properties and corresponding support evidence in the experiments of CDC debiasing method. We have detailed our experiment settings and elucidate their rationality.

● [Reviewer kkqd] We have revised the notation and assumption in the theoretical analysis section to enhance its consistency, thereby rendering both the assumption and the deduction more clear and comprehensible in the theoretical analysis.

● [Reviewer kkqd] We have discussed our contribution, limitation and rationality of 2SLS causal inference to substantiate the credibility of our conclusions.

● [Reviewer S3rs] We have further elaborated on the rationale behind the subjects and methodologies employed in our experiments, which provides additional support for the practicality and generalizability of the conclusions drawn in this paper.

**We hope our response could address all the reviewers' concerns. As the discussion period deadline is approaching, we are more than eager to have further discussions with the reviewers in response to these revisions if there are still any questions.**

Thanks once again for all your valuable comments.

Best Regard,
Submission1134 Authors

---

### Meta-Review · Area_Chair_3k52 · 2024-12-22

**Metareview:**

This manuscript discovers that PLM-based retrievers learn perplexity features for relevance estimation, causing source bias by ranking low-perplexity documents higher. Theoretical analysis shows that the phenomenon stems from the positive correlation between the gradients of the loss functions in the language modeling task and retrieval task. Causal Diagnosis and Correction (CDC), a causal-inspired inference-time debiasing method, is proposed to diagnose the bias effect of perplexity and separate the bias effect from the overall estimated relevance score.

Reviewers commented that the manuscript deals with an important topic, and the theoretical analysis and experiments look sound. During the discussion, some critical corrections and additional clarifications were made. The authors should reflect all these comments and discussions in a revised paper version. In particular, the authors should motivate every hypothesis and make it clear in the paper.

Some remaining concerns include (1) the motivation of each hypothesis not being clearly stated, (2) Reviewer S3rs not being fully convinced about the timeliness, broad interest, and significance of this work.

**Additional Comments On Reviewer Discussion:**

This meta-reviewer recommends accepting this work since the average score, 6.33 (Min: 5, Max: 8), is not too bad. However, I also note that Reviewer S3rs with high confidence (4) gave 5, indicating marginally below the acceptance threshold, saying, "I'm still not fully convinced about the timeliness, broad interest, and significance of this work." Also, Reviewer kkqd raised scores even though the reviewer recognized insufficient explanations about the motivation for the hypothesis.

---

### Decision · Program_Chairs · 2025-01-22

Accept (Poster)